# Low Precision Local Training is Enough for Federated Learning

**Zhiwei Li**[1,*]  **Yiqiu Li**[1,*]  **Binbin Lin**[2,4]  **Zhongming Jin**[3]  **Weizhong Zhang**[1,†]

[1]Fudan University   [2]Zhejiang University   [3]Alibaba Cloud Computing   [4]Fullong Inc.

`{zwli23, yiqiuli22}@m.fudan.edu.cn`    `binbinlin@zju.edu.cn`

`zhongming.jinzm@alibaba-inc.com`    `weizhongzhang@fudan.edu.cn`

## Abstract

Federated Learning (FL) is a prevalent machine learning paradigm designed to address challenges posed by heterogeneous client data while preserving data privacy. Unlike distributed training, it typically orchestrates resource-constrained edge devices to communicate via a low-bandwidth communication network with a central server. This urges the development of more computation and communication efficient training algorithms. In this paper, we propose an efficient FL paradigm, where the local models in the clients are trained with low-precision operations and communicated with the server in low precision format, while only the model aggregation in the server is performed with high-precision computation. We surprisingly find that high precision models can be recovered from the low precision local models with proper aggregation in the server. In this way, both the workload in the client-side and the communication cost can be significantly reduced. We theoretically show that our proposed paradigm can converge to the optimal solution as the training goes on, which demonstrates that low precision local training is enough for FL. Our paradigm can be integrated with existing FL algorithms flexibly. Experiments across extensive benchmarks are conducted to showcase the effectiveness of our proposed method. Notably, the models trained by our method with the precision as low as 8 bits are comparable to those from the full precision training. As a by-product, we show that low precision local training can relieve the over-fitting issue in local training, which under heterogeneous client data can cause the client models drift further away from each other and lead to the failure in model aggregation. Code is released at `https://github.com/digbangbang/LPT-FL`.

## 1  Introduction

Federated learning (FL) [3, 15, 22, 36] is a popular privacy preserving machine learning paradigm to collaboratively learn a global model over the decentralized data. In FL paradigm, the clients are responsible for local training and only have access to their private datasets, while the server plays an essential role in aggregating the clients' updates into a global model. Unlike large-scaled distributed training, FL typically orchestrates resource-constrained edge devices to communicate via a low-bandwidth communication network with a central server. This urges the development of more computation and communication efficient optimization algorithms. The most prevalent approach in FL is developed based on local-SGD [26], which is referred to as FedAvg. In each communication round, the clients individually train their local models for multiple steps and then send them to the server for aggregation. It can be expected that if longer local training process one uses, the greater communication cost saving one can achieve. However, long time local training

---

[*]Equal contribution

[†]Corresponding author: `weizhongzhang@fudan.edu.cn`

could cause the local models drift further away from each other and degrades the aggregated global model's performance or even make the training diverge, especially when the data on the clients are heterogeneous. Therefore, in order to prolong local training processes in FL, extensive efforts have been made in the recent years. For example, the studies [1, 16, 21, 22] modify the local training process by imposing regularization on the client models to enforce them not to drift away from the previous global model. Another line of research [5, 6, 24, 31, 35, 38] focuses on refining the global model in the server aggregation process. These methods typically require a large proxy dataset on the server. Some of them [5, 6, 24] use it to align the outputs of the global model with that of the client ensemble by knowledge distillation. Others develop handcrafted aggregation rules to reweight the updates based on the statistics of updates or performance on proxy data [31, 35, 38] or further tune the global model with proxy data in every communication round [5, 24]. Although promising experimental results have been reported in the literature, it is still unclear that whether there exists more concise and effective FL paradigm, which can reduce both the workload in the client-side and the communication cost.

In this paper, we propose a concise and efficient federated learning paradigm, where the local models in the clients are trained with low precision operations and communicated with the server in low precision format, while only the server-side information integration maintains high-precision computation to ensure the accuracy. Our basic idea is inspired from the Kolmogorov's law [10], that is, the sample average can converge almost surely to the expected value although the samples always contain noise. Therefore, in the server side, we perform the simple moving average on the received low precision models from the clients to recover a high precision global model. In this way, both the workload in the client-side and the communication cost can be significantly reduced. We theoretically proved that our proposed paradigm integrated with FedAVG can converge to the optimal solution as the training goes on, which indicates that low precision local training is enough for federated learning. We extend our method to various existing FL method to show its flexibility. Experiments across extensive benchmarks are conducted to showcase the effectiveness of our proposed method. Notably, the models trained by our method with the precision as low as 8 bits are comparable to those from the full precision federated learning. Compared with some efficient FL designs, our method can achieve significant savings in training memory overhead, and what is more attractive is that our accuracy performance is even better. Our method exhibits another appealing feature in relieving the over-fitting issue in local training. To be precise, in the local training steps, the models can be easily trained to over-fit the local training data as the local dataset is always insufficient. Under heterogeneous client data, it would further cause the client models drift further away from each other and lead to the failure in model aggregation. The experimental results show that our approach can effectively relieve the over-fitting issue since the local training is performed with low precision computation and the expressiveness of the local model is restricted.

Our main contributions are as follows:

- We propose an efficient federated learning paradigm that performs low precision computation during local training, saving computational overhead and communication costs, while being able to restore accuracy through high-precision aggregation on the server side.

- We theoretically proved that the efficient federated learning algorithm we proposed can achieve convergence at a rated of $\mathcal{O}(1/T)$ under certain assumptions for non-iid situations.

- Since the expressiveness of the local model is restricted due to the low precision local training, our approach can relieve the over-fitting issue, which would cause the client models drift further away from each other and lead to the failure in model aggregation when the local dataset are heterogeneous.

- The extensive experimental results demonstrate the effectiveness of our approach. Notably, the models trained by our method with the precision as low as 8 bits are comparable to those from the full precision federated learning.

## 2  Related Work

**Federated Learning.** Fedrated Learning is first proposed by [26] to realize model training without sharing client device data. Many works have continued to solve some challenges of FL such as heterogeneity [16, 22, 25], privacy [2], communication efficiency [11, 18]. Also, some works proposes new FL methods to alleviate data heterogeneity. The vanilla FL method was FedAvg [26].

FedProx [22] utilizes a regularization term while Scaffold [16] sets a control variate to reduce the drift in local training. FedGen [42] and FedFTG [40] maintain a generator, the former is used for local data augmentation, while the latter is used for fine-tuning the server.

**Efficient Federated Learning.** One challenge of FL is the limitation of low bandwidth and computing resources of client devices. [4, 20] assign each client a block mask, resulting in sparse local models. [28] took the transmission speed into consideration and chose the same method as [12] for uploading, uploading compressed gradients. [13] adopts boost training to client-side training overhead. [7, 28] only transmit the trained head to reduce transmission cost. [33] adopts the idea of Network Architecture Searching, e.g., each client selects a sub-network. [14] maintains a series of streamlined models in the server, from which the client selects a tiny model for training. In [9], the client selects a sub-model of the global model for training. In this paper, we address this issue by using low precision local training.

## 3 Preliminary

### 3.1 Federated Learning

Given $N$ clients with their private datasets $\mathcal{D}_k = \{(x_{k,j}, y_{k,j})\}_{j=1}^{|\mathcal{D}_k|}$, $k = 1, \ldots, N$, the optimization objective of FL is always defined as follows:

$$\min_{\mathbf{w}} F(\mathbf{w}) \triangleq \sum_{k=1}^{N} p_k F_k(\mathbf{w}), \tag{1}$$

where $\mathbf{w} \in \mathbb{R}^d$ represents the model parameters, $F_k$ is denoted to be the empirical risk function of client $k$, i.e., $F_k(\mathbf{w}) = \sum_{\xi \in \mathcal{D}_k} \frac{1}{|\mathcal{D}_k|} \ell(\mathbf{w}; \xi)$ with $\ell(\cdot, \cdot)$ being the loss function and $p_k = \frac{|\mathcal{D}_k|}{\sum_{k=1}^{N} |\mathcal{D}_k|}$ denotes the proportion of data contained in client $k$.

FL emphasizes data privacy protection and thus the server is not allowed to access these datasets $\mathcal{D}_k$ directly in model training. The standard method to solve the above training problem of FL is FedAvg [26], which is developed based on local SGD. It is comprised by two steps, i.e., local training on the clients and model aggregation in the server side. The details are presented below.

- In local training, the central server would first randomly select partial clients and broadcasts the latest global model $\mathbf{w}_t$ to them. We denote the selected clients set as $\mathcal{S}_t$ and let $K = |\mathcal{S}_t|$ be the number of selected clients. Then the client $k$ with $k \in \mathcal{S}_t$ would initialize its local model to be $\mathbf{w}_t^k = \mathbf{w}_t$ and then performs local training with $E(\geq 1)$ iterations as follows:

$$\mathbf{w}_{t+1}^k \leftarrow \mathbf{w}_t^k - \eta_t \nabla F_k(\mathbf{w}_t^k; \xi_t^k), k \in \mathcal{S}_t, \tag{2}$$

where $\mathbf{w}_t^k$ is the weights of the $k$-th client in step $t$, $\xi_t^k$ is a mini-batch of samples uniformly chosen from $\mathcal{D}_k$, $\eta_t$ is the step size.

- In model aggregation, FedAvg updates the global model to be the weighted average of the received local models, i.e.,

$$\mathbf{w}_{t+E} \leftarrow \sum_{k \in \mathcal{S}_t} \frac{p_k}{q_t} \mathbf{w}_{t+E}^k, \tag{3}$$

where $q_t = \sum_{k \in \mathcal{S}_t} p_k$ normalize the coefficients.

### 3.2 Block Floating Point Quantization

Fixed point quantization is a standard quantization technique. It uses stochastic rounding to round the numbers up or down at random such that $\mathbb{E}[Q(x)] = x$, where $Q : \mathbb{R} \to \mathbb{R}$ is the quantization function defined as

$$Q(x) = \begin{cases} \text{clip}(\delta \lfloor \frac{x}{\delta} \rfloor, l, u) & \text{w.p. } \lceil \frac{x}{\delta} \rceil - \frac{x}{\delta}, \\ \text{clip}(\delta \lceil \frac{x}{\delta} \rceil, l, u) & \text{w.p. } 1 - (\lceil \frac{x}{\delta} \rceil - \frac{x}{\delta}), \end{cases} \tag{4}$$

here $\text{clip}(x, a, b) = \max(\min(x, b), a)$, $\delta = 2^{-F}$ is the quantization gap represents the distance between successive representable fixed point numbers, $u = 2^{W-F-1} - 2^{-F}$ and $l = -2^{W-F-1}$ represent the upper and lower limits of the representable numbers, respectively. $W$ is the bit width

of quantized numbers, and $F$ is the bit width of quantized numbers' fractional part. In order to improve the utilization efficiency of the bit width and better maintain numerical accuracy when data is unevenly distributed, we choose block floating point quantization. Given a block of numbers $X$, it replaces $\delta = 2^{-F}$ in fixed point quantization to be $\delta = 2^{-(W-2-E(X))}$, where

$$E(X) = \text{clip}(\lfloor \log_2(\max_i |X_i|) \rfloor, -2^{W-F-1}, 2^{W-F-1} - 1). \tag{5}$$

The shared exponent $E(X)$ is usually set to be the largest exponent in $X$ to avoid overflow [37].

## 4    Method

In this section, we introduce our low precision federated learning paradigm. It is comprised of two modules,i.e., one is the low precision local training to reduce the computation and communication cost, the other is the high precision aggregation with moving average to maintain the model accuracy.

### 4.1    Low Precision Local Training

In order to reduce the computation and communication cost, we apply block floating point quantization to all clients' device of local training. The simple version of low precision local training is to convert the local training step in Eqn.(2) into

$$\mathbf{w}_{t+1}^k = Q\big(\mathbf{w}_t^k - \eta_t \nabla F_k(\mathbf{w}_t^k; \xi_t^k)\big). \tag{6}$$

Note that in the above version, we only quantize the updated parameters. We give this version just for the convenience of the theoretical analysis in Section 5. In practice, we quantize the gradient, the activation of each layer, the back-propagation signals, and the momentum in SGD when SGD is adopted as the optimizer. The details are given in Algorithm 1.

---

**Algorithm 1** Low Precision Local Training with All Numbers Quantized

---

**Input:**  Quantization functions $Q_A, Q_E, Q_G, Q_M, Q_W$; Momentum coefficient $\rho$; L layers DNN
$\{f_1, f_2, \ldots, f_L\}$; Loss function $\ell$.
1: **ClientUpdate**$(t, k, w_t^k)$**:**
2:     Get batch $(x_{k,j_t}, y_{k,j_t})$ from $\mathcal{D}_k$
3:     **Forward Propagation:**
4:         $(a_t^k)^{(0)} = x_{k,j_t}$
5:         $(a_t^k)^{(l)} = Q_A(f_l((a_t^k)^{(l-1)}, (w_t^k)^{(l)})), \forall l \in [1, L]$
6:     **Backward Propagation:**
7:         $(e_t^k)^{(L)} = \nabla_{(a_t^k)^{(L)}} \ell((a_t^k)^{(L)}, y_{k,j_t})$
8:         $(e_t^k)^{(l-1)} = Q_E(\frac{\partial f_l((a_t^k)^{(l-1)}, (w_t^k)^{(l)})}{\partial (a_t^k)^{(l-1)}} (e_t^k)^{(l)}), \forall l \in [1, L]$
9:         $(g_t^k)^{(l)} = Q_G(\frac{\partial f_l((a_t^k)^{(l-1)}, (w_t^k)^{(l)})}{\partial (w_t^k)^{(l)}} (e_t^k)^{(l)}), \forall l \in [1, L]$
10:     **Low Precision SGD Update:**
11:         $(v_{t+1}^k)^{(l)} \leftarrow Q_M(\rho(v_t^k)^{(l)} + (g_t^k)^{(l)}), \forall l \in [1, L]$
12:         $(w_{t+1}^k)^{(l)} \leftarrow Q_W((w_t^k)^{(l)} - \eta_t \cdot (v_{t+1}^k)^{(l)}), \forall l \in [1, L]$
13: **Return:** $w_{t+1}^k$

---

### 4.2    High Precision Aggregation

Although low precision training can reduce communication and training overhead, it would lead to a degradation in training accuracy. Inspired from the Kolmogorov's law [10], that is, the sample average can converge almost surely to the expected value although the samples always contain noise, we try to recover high-precision solution from the low-precision local model with a full precision aggregation process. It is implemented with the following two steps:

- Calculate the weighted average of the local models to partially reconvert the precision, i.e.,

$$\mathbf{w}_{t+E} \leftarrow \sum_{k \in \mathcal{S}_t} \frac{p_k}{q_t} \mathbf{w}_{t+E}^k. \tag{7}$$

- Since in most cases clients' data is non-iid and quantization causes error, federated learning is harder to converge, however maintaining a moving average in the server can significantly alleviate the problem. Formally, we denote $\bar{\mathbf{w}}_t$ as the moving average stored in the server, then after aggregation, we update $\bar{\mathbf{w}}_t$ as follow

$$\bar{\mathbf{w}}_t \leftarrow Q(\lambda \bar{\mathbf{w}}_{t-E} + (1 - \lambda)\mathbf{w}_t), \tag{8}$$

where $\lambda$ is a coefficient controlling the influence of current weight. This procedure can further compensate the accuracy degradation due to the low precision local training.

In the next round of local training, the quantized model $Q(\bar{\mathbf{w}}_t)$ will be distributed to the clients to intilize the local models. Our pseudocode in Algorithm 2 depicts the process of low precision local training on the client device and high precision aggregation on the server. In Algorithm 2, $t' = t - E + 1$ and $\mathcal{I}_E = \{nE | n = 1, 2, \cdots\}$ represents the set of global synchronization steps.

---

**Algorithm 2** Federated Learning with Low Precision Local Training

---

1: **Initialize:** $\mathbf{w}_0, \bar{\mathbf{w}}_0 \leftarrow \mathbf{w}_0$
2: **for** $t = 0, 1, \ldots, T - 1$ **do**
3:    **if** $t \equiv 0 \pmod{E}$ **then**
4:        Select $K$ clients from $[N]$ to be $\mathcal{S}_t$
5:        $\mathbf{w}_t^k \leftarrow Q(\bar{\mathbf{w}}_t), k \in \mathcal{S}_t$
6:    **end if**
7:    **for** $k \in \mathcal{S}_t$ **do**
8:        $\mathbf{w}_{t+1}^k \leftarrow Q(\mathbf{w}_t^k - \eta_t \nabla F_k(\mathbf{w}_t^k; \xi_t^k))$ ▷ Client update
9:    **end for**
10:   **if** $t + 1 \in \mathcal{I}_E$ **then**
11:       $\mathbf{w}_{t+1} \leftarrow \sum_{k \in \mathcal{S}_{t'}} \frac{p_k}{q_{t'}} \mathbf{w}_{t+1}^k$ ▷ Server update
12:       $\bar{\mathbf{w}}_{t+1} \leftarrow \lambda \bar{\mathbf{w}}_{t'} + (1 - \lambda)\mathbf{w}_{t+1}$
13:   **end if**
14: **end for**
15: **Return:** $\bar{\mathbf{w}}_T$

---

## 5 Theoretical Analysis

In this section, we give the detailed theoretical results for our low precision FL paradigm. We will first introduce the convergence analysis in the full participation case where all client devices participate (i.e., $K = N$) and then we generalize the results of the analysis to scenarios that are more in line with reality. (i.e., $K < N$). The results demonstrate that we will explore aggregation strategies represented by the FederatedAveraging Algorithm (or FedAvg) and demonstrate that our proposed low precision FL framework can converge to the global optimal solution at a rate of $\mathcal{O}(1/T)$ for non-iid datasets based on strong convexity and smoothness assumptions.

### 5.1 Assumptions and Notations

We need to make necessary assumptions about the objective function on the clients $F_k, k = 1, \ldots, N$. Assumption 1 is about the smoothness and strong convexity of the loss function and Assumption 2 is about the boundness of the gradients. These assumptions are standard and widely adopted in the related studies [41, 29, 30, 39].

**Assumption 1.** $F_1, F_2, ..., F_N$ *are L-smooth and $\mu$-strongly functions, which means that for all* $\mathbf{v}$ *and* $\mathbf{w}$*, the following inequalities hold:*

$$F_k(\mathbf{v}) \leq F_k(\mathbf{w}) + (\mathbf{v} - \mathbf{w})^T \nabla F_k(\mathbf{w}) + \frac{L}{2}\|\mathbf{v} - \mathbf{w}\|_2^2, \qquad (L - smooth) \tag{9}$$

$$F_k(\mathbf{v}) \geq F_k(\mathbf{w}) + (\mathbf{v} - \mathbf{w})^T \nabla F_k(\mathbf{w}) + \frac{\mu}{2}\|\mathbf{v} - \mathbf{w}\|_2^2, \qquad (\mu - strong) \tag{10}$$

*where* $\|\cdot\|^2$ *represents the square of two norms and $k = 1, \ldots, N$.*

**Assumption 2.** *Let $\xi_t^k$ be sample that randomly and uniformly sampled from the local data of the $k$-th client device. For $k = 1, 2, \cdots, N$ and $t = 0, 1, \cdots$, the variance of stochastic gradients in each client device and the expectation of squared two norm of stochastic gradients is bounded:*

$$\mathbb{E}\big\|\nabla F_k(\mathbf{w}_t^k; \xi_t^k) - \nabla F_k(\mathbf{w}_t^k)\big\|_2^2 \leq {\sigma_k}^2, \tag{11}$$

$$\mathbb{E}\big\|\nabla F_k(\mathbf{w}_t^k; \xi_t^k)\big\|_2^2 \leq G^2. \tag{12}$$

**Degree of Data Heterogeneity.** Let $F^*$ denote the global minimum of the objective function $F$, and let $F_k^*$ represent the minimum of the local objective function $F_k$ specific to the $k$-th client. We use the metric $\Gamma$ [23] taking the form of

$$\Gamma = F^* - \sum\nolimits_{k=1}^{N} p_k F_k^*,$$

to measure the degree of heterogeneity of all clients' data distribution. When the data on each client device is iid, as the number of samples increase, $\Gamma$ evidently tends towards zero. However, when faced with non-iid situation, $\Gamma$ tends towards a positive constant and thus it can be used to measure the degree of heterogeneity in the data distribution of each client device.

## 5.2 Convergence Analysis: Full Client Device Participation

First we analyze the convergence of the participation of all clients' device in this section. We integrate our low precision FL framework with FedAvg and train the model for $T$ iterations to obtain the $\bar{\mathbf{w}}_T$, and we expect $T$ to be divided by $E$ so that $\bar{\mathbf{w}}_T$ is the weight after aggregation.

**Theorem 1.** *Under the Assumptions 1 and 2, we set $\kappa = \frac{L}{\mu}$, $\gamma = \max\{8\kappa, E\} - 1$ and $\eta_t = \frac{\mu}{2(t+\gamma)}$. When $t$ satisfing $\delta^2 \leq \eta_t^2 G^2$, low precision FedAvg with full device participation satisfies:*

$$\mathbb{E}[F(\bar{\mathbf{w}}_T)] - F^* \leq \frac{\kappa}{T+\gamma}\left(\frac{2B}{\mu} + \frac{\mu(\gamma+1)}{2}\|\mathbf{w}_1 - \mathbf{w}^*\|_2^2\right), \tag{13}$$

*where*

$$B = 2(\sqrt{d}\delta + 1 + \frac{d}{2})G^2 + 16E^2G^2(2\sqrt{d}\delta + 3) + \frac{1}{N^2}\sum\nolimits_{k=1}^{N}{\sigma_k}^2 + 6L\Gamma. \tag{14}$$

## 5.3 Convergence Analysis: Partial Client Device Participation

Compared to full participation situation, the partial participation situation is more in line with the reality. We need to make more assumption on how to choose $\mathcal{S}_t$.

**Assumption 3.** *Assume $\mathcal{S}_t$ contains a subset of $K$ indices uniformly sampled from $[N]$ without replacement. In addition, the data is balanced in the sence that $p_1 = \cdots = p_N = \frac{1}{N}$. The aggregation step of FedAvg performs $\mathbf{w}_{t+E} \leftarrow \frac{1}{K}\sum_{k\in\mathcal{S}_t}\mathbf{w}_{t+E}^k$*

**Theorem 2.** *Under the Assumptions 1 to 3, we choose $\kappa = \frac{L}{\mu}$, $\gamma = \max\{8\kappa, E\} - 1$, $\eta_t = \frac{\mu}{2(t+\gamma)}$ and $B = 2(\sqrt{d}\delta + 1 + \frac{d}{2})G^2 + 16E^2G^2(2\sqrt{d}\delta + 3) + \frac{1}{N^2}\sum_{k=1}^{N}{\sigma_k}^2 + 6L\Gamma$, $C = \frac{N-K}{N-1}\frac{8}{K}E^2G^2(2\sqrt{d}\delta + 3) + dG^2$. When $t$ satisfing $\delta^2 \leq \eta_t^2 G^2$, low precision FedAvg with partial client device participation satisfies:*

$$\mathbb{E}[F(\bar{\mathbf{w}}_T)] - F^* \leq \frac{\kappa}{T+\gamma}\left(\frac{2(B+C)}{\mu} + \frac{\mu(\gamma+1)}{2}\|\mathbf{w}_1 - \mathbf{w}^*\|_2^2\right). \tag{15}$$

# 6 Experiments

In this section, we conduct extensive experiments to verify the effectiveness of our methods in the following five aspects:

- When integrated with FedAvg, the models trained by our method with the low precision are comparable to (if not better than) those from the full precision training. This would also verify our theoretical results (Section 6.1).
- Our method can effectively relieve the over-fitting issue in FL. See Section (Section 6.2).

- Our paradigm can be integrated with existing FL methods flexibly and preserve the performance even with a low precision local training (Section 6.3).
- Ablation studies on the effectiveness of moving average in model aggregation and the transferability over various neural networks (Section 6.4).
- Comparison with other efficient FL techniques (Section 6.5).

**Remark 1.** *Similar with the existing low precision training studies [32], We do not give the results on the running time to show the real acceleration. The reason is that to achieve real acceleration, we need to implement our method integrated with the professional hardware. Moreover, such implementation is standard for the professional hardware platforms.*

**Benchmark Datasets and Baseline.** We conduct experiments over four commonly used datasets: FashionMnist [34], CIFAR10 [19], CIFAR100 [19] and CINIC10 [8]. Four commonly used FL methods: 1) FedAvg [26]; 2) regularization-based strategy FedProx [22]; 3) data-dependent knowledge distillation strategy ABAvg [35] 4) data-free knowledge distillation strategy FedFTG [40] and FedGen [42] are adopted as the baselines.

**Configurations.** We follow the configurations in the recent studies [26, 42, 27] for fair comparison. To be precise, for FashionMNIST, CIFAR10, CINIC10 and CIFAR100, we run 200 communication rounds with local epoch set to 1. There are 80 clients in total, and the participation ratio in each round is set to 40%. We use Dirichlet distribution to simulate non-iid data distribution and set $\alpha$ to 0.01, 0.04, and 0.16. The smaller $\alpha$ is, the more serious the data heterogeneity is. For the network choice, we use ConvNet following with 3 layers, and the hidden dimension is set to 128. The local learning rate is set to $10^{-3}$ with Adam optimizer [17]. We report the last 5 round global model's average performance evaluated using the test split of the datasets. For quantization method, we adopt the Block FLoating Point Quantization with the number of bits used set to 6, 8 and 32 (without quantization). Some of the other hyperparameter settings are included in the Appendix C.

## 6.1 Results on FedAvg

We demonstrate the superior performance of our Low Precision FL method with FedAvg by conducting experiments over comprehensive datasets, various precision and heterogeneity values $\alpha$.

**Heterogeneity.** As shown in Table 1, it is as expected that when the heterogeneity goes higher, that is, when $\alpha$ decreases, the server performance worsens. Nevertheless, our proposed method can always maintain or improve the performance of the original case (bits = 32, w/o. avg) when using 8 quantizaiton bits with moving average, which empirically validates the effectiveness of our proposed method.

**Quantization Bits.** We conduct experiments on 3 quantization bits: 32, 8, 6 (shown in Table 1, Figure 1), as we observe that in most cases 8 bits is enough to hold the performance of full-precision and when the used bits is 6, the server performance begins to decrease due to the low precision level.

Table 1: Results of our method integrated with FedAvg over various levels of heterogeneity and precision. The results with moving average demonstrate that our method can match the performance of full-precision federated learning even with all numbers quantized down to 8 bits. The results in the bottom three rows indicates the without moving average, training with low precision would lead to performance degradation.

| Averaging | Prec. | $\alpha = 0.01$ | | | | $\alpha = 0.04$ | | | | $\alpha = 0.16$ | | | |
|---|---|---|---|---|---|---|---|---|---|---|---|---|---|
| | | FMNIST | CIFAR10 | CINIC10 | CIFAR100 | FMNIST | CIFAR10 | CINIC10 | CIFAR100 | FMNIST | CIFAR10 | CINIC10 | CIFAR100 |
| | 8 bit | $80.1 \pm 0.7$ | $53.3 \pm 2.7$ | $43.3 \pm 1.2$ | $15.2 \pm 0.2$ | $83.8 \pm 0.1$ | $57.4 \pm 0.6$ | $50.8 \pm 2.2$ | $34.1 \pm 0.4$ | $90.6 \pm 0.3$ | $72.4 \pm 0.7$ | $58.2 \pm 1.5$ | $42.9 \pm 0.1$ |
| *w avg* | 6 bit | $78.0 \pm 1.4$ | $49.3 \pm 0.9$ | $39.2 \pm 1.5$ | $12.4 \pm 0.0$ | $81.8 \pm 0.1$ | $53.6 \pm 0.6$ | $46.8 \pm 1.6$ | $22.4 \pm 0.3$ | $89.7 \pm 0.5$ | $69.6 \pm 0.7$ | $56.8 \pm 0.3$ | $33.8 \pm 0.0$ |
| | 32 bit | $80.7 \pm 0.6$ | $52.7 \pm 0.6$ | $41.0 \pm 0.8$ | $15.6 \pm 0.0$ | $83.6 \pm 0.0$ | $58.6 \pm 0.3$ | $49.7 \pm 1.8$ | $30.7 \pm 0.3$ | $90.7 \pm 0.2$ | $73.0 \pm 0.9$ | $58.6 \pm 0.9$ | $41.7 \pm 0.3$ |
| | 8 bit | $72.2 \pm 2.0$ | $25.6 \pm 1.4$ | $20.0 \pm 1.2$ | $7.32 \pm 0.4$ | $79.3 \pm 1.2$ | $45.5 \pm 1.6$ | $38.3 \pm 1.9$ | $14.7 \pm 2.1$ | $87.4 \pm 2.2$ | $63.6 \pm 0.8$ | $47.5 \pm 1.9$ | $25.6 \pm 0.4$ |
| *w/o avg* | 6 bit | $41.5 \pm 4.8$ | $20.8 \pm 0.8$ | $18.1 \pm 1.0$ | $3.16 \pm 0.7$ | $72.5 \pm 1.7$ | $30.8 \pm 1.3$ | $28.8 \pm 1.5$ | $6.75 \pm 0.1$ | $83.0 \pm 1.3$ | $44.6 \pm 1.1$ | $39.3 \pm 1.4$ | $9.62 \pm 0.2$ |
| | 32 bit | $79.5 \pm 2.7$ | $41.1 \pm 2.4$ | $35.4 \pm 4.1$ | $12.5 \pm 0.4$ | $83.1 \pm 2.1$ | $54.4 \pm 2.0$ | $43.2 \pm 3.6$ | $28.2 \pm 0.5$ | $90.2 \pm 0.5$ | $71.9 \pm 1.5$ | $57.0 \pm 1.3$ | $39.1 \pm 0.4$ |

## 6.2 Quantization Relieves Overfitting

We present the averaged local training loss and the global testing loss over training in Figure 2. It can be seen that our method can effectively reduce testing losses on the server. The commonality

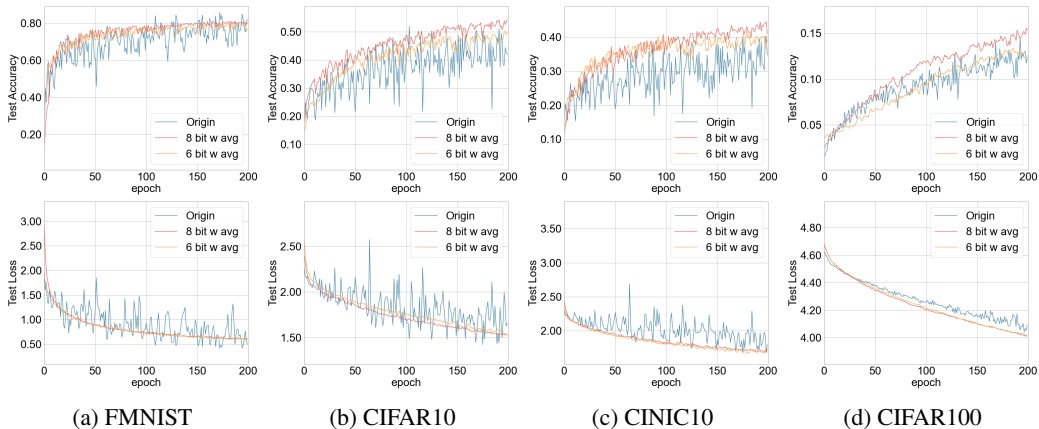

(a) FMNIST      (b) CIFAR10      (c) CINIC10      (d) CIFAR100

Figure 1: Accuracy and loss of FedAvg with full precision (origin), our method with precsion levels of 8 bit and 6 bit. We set $\alpha = 0.01$ on all the four datasets. Our method exhibits an effective reduction in fluctuation variance and improves the stability of training. The reason is that compared with the full precision training, our low precision local training can prevent the client models to drift further away from each other and overfit the local datasets, making the aggregation stable.

with the original method is that when each round of communication starts retraining, the training loss of the client will be greatly increased due to the heterogeneity of the data $\mathcal{D}_k$. Subsequently, due to the highly imbalanced local data categories, the model quickly reached an overfitting state. It can be seen that at the beginning of each training round, under our method, the customer's training loss will not exceed the original training loss. At the same time, with some training steps, our training loss remains above the original training loss, which means our method can alleviate the overfitting problem of local training.

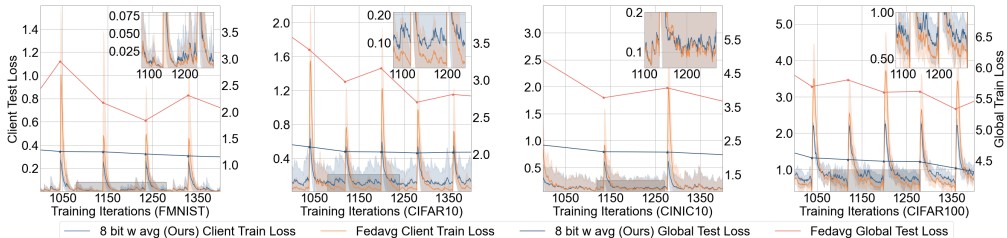

Figure 2: Effectiveness of our method in relieving the over-fitting issue. We present the averaged local **training** loss and the global **test** loss over training. We select a part of the training procedure (iteration 1000 to 1400) for display, and enlarge a part of the picture in the upper right corner to show more details. We can observe that the local training loss of FedAvg (full precision) is siginifcantly lower than our method, however its global test loss is much higher than us and fluctuates dramatically.

## 6.3 Results on Other FL Methods

The four FL methods we used are each representative. ABAvg and FedFTG are similar to FedAvg in local training, but the former only performs weight adjustment on the server side, while the latter uses knowledge distillation to fine-tune the server. FedProx and FedGen are similar to FedAvg in the server side, but the former only has regularization constraints on local training, while the latter uses the generator for regularization adjustment in local training. As is demonstrated in Table 2, we can see that, regardless of the FL method chosen, our low precision FL algorithm has a significant improvement in prediction accuracy compared to the original FL method, especially in dataset CIFAR10, CINIC10 and CIFAR100.

Table 2: The results of our method when integrated with existing FL methods ABAvg, FedProx, FedGen and FedFTG. Our method can improve the effectiveness of FL with a certain training cost, and can be applied in both regularization and distillation. The complete result is given in Table 4. It shows that integrated with our low precision training technique, these existing methods with the precision of 8 bits can match (or even better than) the performance with full precision.

| Method | Averaging | Prec. | α = 0.01 | | | | α = 0.04 | | | | α = 0.16 | | | |
|---|---|---|---|---|---|---|---|---|---|---|---|---|---|---|
| | | | FMNIST | CIFAR10 | CINIC10 | CIFAR100 | FMNIST | CIFAR10 | CINIC10 | CIFAR100 | FMNIST | CIFAR10 | CINIC10 | CIFAR100 |
| ABAvg | w avg | 8 bit | 80.7 ± 0.4 | 51.1 ± 1.4 | 41.4 ± 0.8 | 22.4 ± 0.9 | 88.4 ± 1.2 | 55.8 ± 0.5 | 50.9 ± 2.7 | 34.3 ± 0.2 | 90.2 ± 0.6 | 72.3 ± 1.0 | 58.2 ± 1.4 | 42.5 ± 0.3 |
| | w/o avg | 32 bit | 78.4 ± 2.8 | 44.5 ± 4.2 | 26.4 ± 3.8 | 16.0 ± 0.4 | 83.5 ± 2.3 | 56.1 ± 3.1 | 42.1 ± 3.8 | 27.3 ± 0.6 | 90.3 ± 1.0 | 71.3 ± 1.3 | 58.1 ± 1.6 | 39.9 ± 0.7 |
| FedProx | w avg | 8 bit | 83.0 ± 0.6 | 51.4 ± 0.5 | 42.2 ± 1.4 | 18.2 ± 0.1 | 86.5 ± 1.2 | 63.2 ± 0.3 | 48.4 ± 2.4 | 35.5 ± 0.5 | 88.8 ± 0.7 | 71.9 ± 1.0 | 57.6 ± 0.6 | 43.7 ± 0.1 |
| | w/o avg | 32 bit | 81.1 ± 3.8 | 45.1 ± 3.4 | 32.2 ± 3.6 | 16.8 ± 0.7 | 86.3 ± 1.5 | 57.0 ± 2.1 | 46.0 ± 2.0 | 29.3 ± 1.3 | 89.1 ± 0.4 | 72.5 ± 0.5 | 58.0 ± 0.7 | 43.2 ± 0.7 |
| FedGen | w avg | 8 bit | 79.9 ± 2.2 | 45.5 ± 1.9 | 34.7 ± 2.1 | 20.1 ± 1.1 | 86.2 ± 0.7 | 60.5 ± 1.4 | 47.4 ± 1.3 | 32.9 ± 0.7 | 91.1 ± 0.3 | 74.0 ± 0.9 | 58.4 ± 1.0 | 43.2 ± 0.2 |
| | w/o avg | 32 bit | 80.2 ± 2.6 | 43.1 ± 4.3 | 30.0 ± 4.8 | 15.6 ± 1.3 | 83.7 ± 0.8 | 54.1 ± 2.3 | 45.3 ± 2.3 | 29.8 ± 0.3 | 90.3 ± 0.3 | 72.4 ± 1.0 | 57.3 ± 0.3 | 40.7 ± 0.6 |
| FedFTG | w avg | 8 bit | 82.0 ± 1.1 | 52.8 ± 0.7 | 39.6 ± 0.9 | 22.9 ± 0.8 | 87.7 ± 0.2 | 59.1 ± 0.2 | 51.2 ± 0.3 | 35.8 ± 0.2 | 90.9 ± 0.3 | 73.7 ± 0.9 | 58.7 ± 1.7 | 42.9 ± 0.3 |
| | w/o avg | 32 bit | 80.6 ± 2.2 | 46.1 ± 7.3 | 32.4 ± 1.0 | 15.9 ± 1.9 | 84.6 ± 1.7 | 58.9 ± 4.1 | 47.3 ± 1.9 | 28.3 ± 0.4 | 90.7 ± 0.4 | 73.8 ± 1.0 | 57.9 ± 0.3 | 41.6 ± 0.6 |

## 6.4 Ablation Study

**Moving Average.** We conduct ablation study to validate the effectiveness of the moving average for the server, from Table 1, the average operation has a significant effect on the server performance over various settings. It is consistent of our expectation that averaging can alleviate the error introduced by quantization and the error by heterogeneity.

**Architecture Generalization.** We use another network structure, MLP, to verify that our method can be applied to other networks. The MLP was set to 3 layers, with the hidden dimension of each layer being 128, and experiments were conducted on FashionMNIST with $\alpha$ of 0.01 and 0.04. From Table 3, our method can still maintain the accuracy, which also demonstrates its generalization ability on other network structures.

Table 3: To display our method can generalize to other architectures, the same experiments were also performed on the MLP, using the FashionMNIST, setting $\alpha = 0.01$ and $\alpha = 0.04$. Through comparison, it is found that the accuracy under MLP is slightly lower than that of ConvNet. However, our method can still maintain or improve the accuracy on MLP.

| | Method | 8 bit w avg | | 6 bit w avg | | 32 bit w avg | | 32 bit w/o avg | |
|---|---|---|---|---|---|---|---|---|---|
| | | MLP | ConvNet | MLP | ConvNet | MLP | ConvNet | MLP | ConvNet |
| | FedAvg | 73.4 ± 1.2 | 80.1 ± 0.7 | 72.5 ± 0.2 | 78.0 ± 1.4 | 74.1 ± 1.0 | 80.7 ± 0.6 | 62.1 ± 4.1 | 79.5 ± 2.7 |
| | ABAvg | 75.1 ± 0.4 | 80.7 ± 0.4 | 69.7 ± 1.9 | 78.0 ± 0.7 | 75.8 ± 0.7 | 79.5 ± 1.1 | 58.6 ± 6.9 | 78.4 ± 2.8 |
| $\alpha = 0.01$ | FedProx | 74.6 ± 0.5 | 83.0 ± 0.6 | 73.6 ± 0.4 | 78.1 ± 0.4 | 74.7 ± 0.4 | 82.9 ± 0.9 | 70.9 ± 4.3 | 81.1 ± 3.8 |
| | FedGen | 73.3 ± 1.1 | 79.9 ± 2.2 | 70.9 ± 0.4 | 78.5 ± 1.3 | 73.4 ± 1.5 | 78.9 ± 1.5 | 72.2 ± 3.8 | 80.2 ± 2.6 |
| | FedFTG | 78.1 ± 0.8 | 82.0 ± 1.1 | 74.8 ± 1.2 | 80.0 ± 0.4 | 77.1 ± 1.9 | 82.0 ± 0.7 | 63.1 ± 1.3 | 80.6 ± 2.2 |
| | FedAvg | 79.5 ± 0.1 | 83.8 ± 0.2 | 78.4 ± 0.3 | 81.8 ± 0.1 | 79.1 ± 0.2 | 83.6 ± 0.0 | 78.8 ± 1.7 | 83.1 ± 2.1 |
| | ABAvg | 79.2 ± 0.2 | 88.4 ± 1.2 | 78.6 ± 0.1 | 86.1 ± 0.5 | 79.1 ± 0.2 | 88.4 ± 0.8 | 77.2 ± 3.7 | 83.5 ± 2.3 |
| $\alpha = 0.04$ | FedProx | 81.7 ± 0.4 | 86.5 ± 1.2 | 79.9 ± 0.3 | 79.9 ± 0.2 | 81.9 ± 0.5 | 86.9 ± 0.6 | 80.8 ± 2.4 | 86.3 ± 1.5 |
| | FedGen | 79.0 ± 0.5 | 86.2 ± 0.7 | 76.7 ± 0.7 | 81.9 ± 1.3 | 79.5 ± 1.0 | 83.6 ± 1.2 | 80.0 ± 0.6 | 83.7 ± 0.8 |
| | FedFTG | 83.5 ± 0.5 | 87.7 ± 0.2 | 79.2 ± 0.6 | 81.9 ± 0.3 | 84.0 ± 0.6 | 87.6 ± 0.6 | 75.5 ± 4.9 | 84.6 ± 1.7 |

## 6.5 Versus Efficient FL Method

In order to demonstrate that our method can effectively reduce computational costs and memory consumption, we compared it with two personalized model compression FL algorithms. HeteroFL [9] and Split-Mix [14] can both achieve the purpose of reducing training overhead by using smaller models for local training. We use the ConvNet with the same hidden dimension to conduct experiments, using CIFAR10 in the process, $\alpha = 0.01$. SplitMix uses a split ratio of 1/8 by default. We re-tested at a split ratio of 1/16 to achieve the same compression ratio as HeteroFL. In Figure 3a, the vertical dotted line indicates the compression ratio of HeteroFL, more details are in the Appendix E. Compared with methods that directly compress model parameters, our method can achieve higher accuracy with lower memory consumption. Because quantization does not bring about the deletion of model parameters, there will be no aggregation problems when the model compression rate is too high.

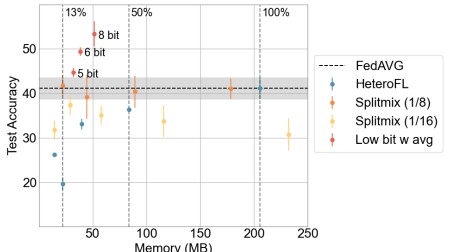

(a) Training cost Versus Acc.

| Method | Type | Acc | Communication cost | Train cost |
|---|---|---|---|---|
| HeteroFL | (1) | $41.1 \pm 2.4$ | 39.1 | 205.4 |
| | (1/2) | $36.4 \pm 0.5$ | 19.6 | 83.7 |
| | (1/4, 1/8) | $29.2 \pm 2.1$ | 7.3 | 31.1 |
| SplitMix | (1) | $41.0 \pm 2.3$ | 39.1 | 178.4 |
| | (1/2) | $40.4 \pm 3.6$ | 19.6 | 89.2 |
| | (1/4, 1/8) | $38.9 \pm 3.5$ | 7.3 | 33.4 |
| FedAVG | 8 $bit$ | $53.3 \pm 2.8$ | 9.8 | 51.3 |
| | 6 $bit$ | $49.4 \pm 1.0$ | 7.3 | 38.5 |
| | 5 $bit$ | $44.7 \pm 1.0$ | 6.1 | 32.1 |

(b) Results on CIFAR10 with $\alpha = 0.01$.

Figure 3: Results on CIFAR10 with $\alpha = 0.01$. ( ) denotes the percentage of models on the clients. We use the number of weights, activation, and gradients of local training to approximate the training cost (MB / client) and communication cost (MB / round).

# 7 Conclusion

In this paper, we propose an efficient FL paradigm, where the local models in the clients are trained with low-precision operations and communicated with the server in low precision format, while only the model aggregation in the server is performed with high-precision computation. We theoretically show that our proposed paradigm can converge to the optimal solution as the training goes on, which demonstrates that low precision local training is enough for FL. Our paradigm can be integrated with existing FL algorithms flexibly. Experiments across extensive benchmarks are conducted to showcase the effectiveness of our proposed method. As a by-product, we show that low precision local training can relieve the over-fitting issue in local training.

# 8 Acknowledgements

Authors acknowledge the support in part by The National Nature Science Foundation of China grant No: 62472097, The National Nature Science Foundation of China grant No: 62273303, Yongjiang Talent Introduction Programme grant No: 2022A-240-G.

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

This appendix can be divided into several parts. To be precise:

- Section A gives the detailed proof for Theorem 1.
- Section B gives the detailed proof for Theorem 2. Compared to Theorem 1, Theorem 2 assumes that only a subset of client devices participate in the training, which is more in line with real-world scenarios.
- Section C introduce four FL methods'(ABAvg, FedProx, FedGen, FedFTG) hyperparameters setting in our experiments.
- Section D provides detailed experiment results of four FL methods(ABAvg, FedProx, FedGen, FedFTG).
- Section E presents the comparsion between our low precision FL method and two efficient FL method in training overhead.
- Section F gives Limitation of this paper.
- Section G gives Broader Impacts of this paper.

## A    Proofs of Theorem 1

We analyze our low-precision federated learning algorithm in the setting of full device participation in this section. For the sake of convenience in proving, let's take the value of $p_k$ to be $\frac{1}{N}$, disregard the moving average, assume that quantization is only applied to model parameters and quantization gap is always $\delta$. Let $\mathcal{I}_E$ be a set composed of the sequence number of global aggregation steps, i.e., $\mathcal{I}_E = \{nE|n = 1, 2, \cdots\}$. Suppose the model weights are updated as follows:

$$\mathbf{v}_{t+1}^k = Q(\mathbf{w}_t^k - \eta_t \nabla F_k(\mathbf{w}_t^k; \xi_t^k)).$$

$$\mathbf{w}_{t+1}^k = \begin{cases} \mathbf{v}_{t+1}^k & \text{if } t+1 \notin \mathcal{I}_E, \\ Q(\frac{\sum_{k=1}^N \mathbf{v}_{t+1}^k}{N}) & \text{if } t+1 \in \mathcal{I}_E. \end{cases}$$

where $\mathbf{w}_t^k$ is an $d$-dimensional vector, which represents the weight of the $k$-th client device in step $t$. $F_k$ represents the objective function of the $k$-th device. $N$ is the total number of clients device participating in the training, and $T$ is the total number of steps trained, $\xi_t^k$ represents a batch of data randomly sampled from the $k$-th client device's local dataset $\mathcal{D}_k$ in step $t$, $\nabla F_k(\mathbf{w}_t^k; \xi_t^k)$ is stochastic gradient, $\eta_t$ represents learning rate, $Q$ represents the quantization function, and its formula has already been given in (4).

**Notations.** To further simplify our proof process, we introduce the following additional definitions:

$$\bar{\mathbf{w}}_0 = \mathbf{w}_0, \bar{\mathbf{w}}_t = \frac{1}{N}\sum_{k=1}^N \mathbf{w}_t^k, \bar{\mathbf{v}}_t = \frac{1}{N}\sum_{k=1}^N \mathbf{v}_t^k, \bar{\mathbf{g}}_t = \frac{1}{N}\sum_{k=1}^N \nabla F_k(\mathbf{w}_t^k), \mathbf{g}_t = \frac{1}{N}\sum_{k=1}^N \nabla F_k(\mathbf{w}_t^k, \xi_t^k).$$

$\mathbf{w}_t$ represents the weight of the global model in step $t$. Obviously, we can deduce $\mathbb{E}\mathbf{g}_t = \bar{\mathbf{g}}_t$ and $\bar{\mathbf{v}}_{t+1} = \bar{\mathbf{w}}_t - \eta_t \mathbf{g}_t$.

**Lemma 1.** *If $w$ is a scalar and can be written as $w = \tilde{w} + b\delta$, satisfying $-\delta < \tilde{w} < \delta$, $\delta$ is the quantization gap, represents the distance between successive representable fixed-point numbers, then we have:*

$$Q(w) - w = \begin{cases} -\tilde{w}, & w.p. \quad 1 - sign(\tilde{w})\frac{\tilde{w}}{\delta}, \\ sign(\tilde{w})\delta - \tilde{w}, & w.p. \quad sign(\tilde{w})\frac{\tilde{w}}{\delta}. \end{cases}$$

*where $Q(\cdot)$ is quantization function. And we always can find a suitable $\tilde{w}$ that satisfies $|\tilde{w}| \le |w|$ .*

*Proof.* The quantization function is:

$$Q(w) = \begin{cases} clip(\delta\lfloor\frac{w}{\delta}\rfloor, l, u), & w.p. \quad \lceil\frac{w}{\delta}\rceil - \frac{w}{\delta}, \\ clip(\delta\lceil\frac{w}{\delta}\rceil, l, u), & w.p. \quad 1 - \lceil\frac{w}{\delta}\rceil + \frac{w}{\delta}. \end{cases}$$

When $\tilde{w} > 0$:

$$Q(w) - w = \begin{cases} -\tilde{w}, & w.p. \ 1 - \frac{\tilde{w}}{\delta}, \\ \delta\tilde{w}, & w.p. \ \frac{\tilde{w}}{\delta}. \end{cases}$$

When $\tilde{w} < 0$:

$$Q(w) - w = \begin{cases} -\delta - \tilde{w}, & w.p. \ -\frac{\tilde{w}}{\delta}, \\ -\tilde{w}, & w.p. \ 1 - \frac{\tilde{w}}{\delta}. \end{cases}$$

So we have:

$$Q(w) - w = \begin{cases} -\tilde{w}, & w.p. \ 1 - sign(\tilde{w})\frac{\tilde{w}}{\delta}, \\ sign(\tilde{w})\delta - \tilde{w}, & w.p. \ sign(\tilde{w})\frac{\tilde{w}}{\delta}. \end{cases}$$

Next we prove that we always can find a suitable $\tilde{w}$ that satisfies $|\tilde{w}| \leq |w|$. When $w \geq \delta$, we can choose a $\tilde{w}$ satisfying $\tilde{w} \geq 0$, then we can have $|\tilde{w}| \leq |w|$. When $w \leq -\delta$, we can choose a $\tilde{w}$ satisfying $\tilde{w} \leq 0$, then we also can have $|\tilde{w}| \leq |w|$. When $-\delta < w < \delta$, we can choose a $\tilde{w}$ satisfying $\tilde{w} = w$, then we also can have $|\tilde{w}| \leq |w|$. In summary, we always can find a suitable $\tilde{w}$ that satisfies $|\tilde{w}| \leq |w|$. $\qquad\square$

**Discussion.** In the lemma 2 below, we analyze the convergence of the algorithm under the condition that $\delta^2 \leq \eta_t^2 G^2$, where $G^2$ is given in assumption 2 and represents the constraint on the expected value of the squared two-norm of the stochastic gradient. The right side of the inequality indicates the change in the iterative parameters during the gradient descent process, that is, the size of the gradient multiplied by the size of the learning rate. If the magnitude of this value is already less than the quantization precision, it will cause the gradient descent to fail, and the model parameters will not continue to change. In this case, it makes no sense to continue optimization, so analyzing convergence under this condition is practical.

**Lemma 2.** *We assume that $\eta_t \leq \frac{1}{4L}$, based on assumption 1 and 2, when t satisfying $\delta^2 \leq \eta_t^2 G^2$ we have:*

$$\mathbb{E}\left\|\bar{\mathbf{w}}_{t+1} - \mathbf{w}^*\right\|_2^2 \leq 2(\sqrt{d}\delta + 1 + \frac{d}{2})\eta_t^2 G^2 + (1 - \eta_t\mu)\mathbb{E}\left\|\bar{\mathbf{w}}_t - \mathbf{w}^*\right\|_2^2 + 6L\eta_t^2\Gamma$$

$$+ \frac{2}{N}\sum_{K=1}^{N}\left\|\bar{\mathbf{w}}_t - \mathbf{w}_t^k\right\|_2^2 + \mathbb{E}\eta_t^2\left\|\mathbf{g}_t - \bar{\mathbf{g}}_t\right\|_2^2.$$

*Proof.* When $t + 1 \notin \mathcal{I}_E$, we can easily derive that $\bar{\mathbf{w}}_{t+1} = \bar{\mathbf{v}}_{t+1}$. So we have:

$$\left\|\bar{\mathbf{w}}_{t+1} - \mathbf{w}^*\right\|_2^2$$

$$= \left\|\bar{\mathbf{v}}_{t+1} - \mathbf{w}^*\right\|_2^2$$

$$= \left\|\frac{1}{N}\sum_{k=1}^{N}Q(\mathbf{w}_t^k - \eta_t\nabla F_k(\mathbf{w}_t^k, \xi_t^k)) - \mathbf{w}^*\right\|_2^2$$

$$= \left\|\bar{\mathbf{w}}_t - \eta_t\mathbf{g}_t - \mathbf{w}^* + \frac{1}{N}\sum_{k=1}^{N}Q(\mathbf{w}_t^k - \eta_t\nabla F_k(\mathbf{w}_t^k, \xi_t^k)) - \frac{1}{N}\sum_{k=1}^{N}(\mathbf{w}_t^k - \eta_t\nabla F_k(\mathbf{w}_t^k, \xi_t^k))\right\|_2^2$$

$$= \underbrace{\left\|\bar{\mathbf{w}}_t - \eta_t\mathbf{g}_t - \mathbf{w}^*\right\|_2^2}_{A_1} + \underbrace{\left\|\frac{1}{N}\sum_{k=1}^{N}(Q(\mathbf{w}_t^k - \eta_t\nabla F_k(\mathbf{w}_t^k, \xi_t^k)) - (\mathbf{w}_t^k - \eta_t\nabla F_k(\mathbf{w}_t^k, \xi_t^k)))\right\|_2^2}_{A_2}$$

$$+ \underbrace{2\langle\bar{\mathbf{w}}_t - \eta_t\mathbf{g}_t - \mathbf{w}^*, \frac{1}{N}\sum_{k=1}^{N}(Q(\mathbf{w}_t^k - \eta_t\nabla F_k(\mathbf{w}_t^k, \xi_t^k)) - (\mathbf{w}_t^k - \eta_t\nabla F_k(\mathbf{w}_t^k, \xi_t^k)))\rangle}_{A_3}.$$

$\langle\cdot\rangle$ represents inner product operation. Due to $\mathbb{E}Q(x) = x$, according to the Law of iterated expectations, $\mathbb{E}A_3 = 0$. Next we aim to bound $A_2$. $\mathbb{E}A_2$ is:

$$\mathbb{E}\left\|\frac{1}{N}\sum_{k=1}^{N}(Q(\mathbf{w}_t^k - \eta_t\nabla F_k(\mathbf{w}_t^k, \xi_t^k)) - (\mathbf{w}_t^k - \eta_t\nabla F_k(\mathbf{w}_t^k, \xi_t^k)))\right\|_2^2$$

$$= \frac{1}{N^2}\sum_{k=1}^{N}\mathbb{E}(\left\|Q(\mathbf{w}_t^k - \eta_t\nabla F_k(\mathbf{w}_t^k, \xi_t^k)) - (\mathbf{w}_t^k - \eta_t\nabla F_k(\mathbf{w}_t^k, \xi_t^k))\right\|_2^2$$

$$+ \frac{1}{N^2}\sum_{k_1\neq k_2}\mathbb{E}\langle Q(\mathbf{w}_t^{k_1} - \eta_t\nabla F_{k_1}(\mathbf{w}_t^{k_1}, \xi_t^{k_1})) - (\mathbf{w}_t^{k_1} - \eta_t\nabla F_{k_1}(\mathbf{w}_t^{k_1}, \xi_t^{k_1})),$$

$$Q(\mathbf{w}_t^{k_2} - \eta_t\nabla F_{k_2}(\mathbf{w}_t^{k_2}, \xi_t^{k_2})) - (\mathbf{w}_t^{k_2} - \eta_t\nabla F_{k_2}(\mathbf{w}_t^{k_2}, \xi_t^{k_2})))\rangle$$

$$= \frac{1}{N^2}\sum_{k=1}^{N}\mathbf{E}\left\|Q(\mathbf{w}_t^k - \eta_t\nabla F_k(\mathbf{w}_t^k, \xi_t^k)) - (\mathbf{w}_t^k - \eta_t\nabla F_k(\mathbf{w}_t^k, \xi_t^k))\right\|_2^2.$$

In the above equation, we used the law of iterated expectations.

The quantization function is:

$$Q(w) = \begin{cases} clip(\delta\lfloor\frac{w}{\delta}\rfloor, l, u), & w.p. \quad \lceil\frac{w}{\delta}\rceil - \frac{w}{\delta}, \\ clip(\delta\lceil\frac{w}{\delta}\rceil, l, u), & w.p. \quad 1 - \lceil\frac{w}{\delta}\rceil + \frac{w}{\delta}. \end{cases}$$

$\mathbf{w}_t^k$ is a quantified vector, therefore, its p-th component $(\mathbf{w}_t^k)_p = c\,\delta$, c is an integer. We rewrite $\left[\eta_t\nabla F_k(\mathbf{w}_t^k, \xi_t^k)\right]_p = b\delta + \tilde{w}$. Where b is an integer and $|\tilde{w}| \leq |w|$. So that we can have:

$$[Q(\mathbf{w}_t^k - \eta_t\nabla F_k(\mathbf{w}_t^k, \xi_t^k)) - (\mathbf{w}_t^k - \eta_t\nabla F_k(\mathbf{w}_t^k, \xi_t^k))]_p$$

$$= \begin{cases} -sign(\tilde{w})\delta + \tilde{w} & w.p. \quad \frac{sign(\tilde{w})\tilde{w}}{\delta}, \\ \tilde{w} & w.p. \quad 1 - \frac{sign(\tilde{w})\tilde{w}}{\delta}. \end{cases}$$

The expectation after the square is:

$$\mathbb{E}[Q(\mathbf{w}_t^k - \eta_t\nabla F_k(\mathbf{w}_t^k, \xi_t^k)) - (\mathbf{w}_t^k - \eta_t\nabla F_k(\mathbf{w}_t^k, \xi_t^k))]_p^2$$

$$= \tilde{w}^2\left[1 - \frac{sign(\tilde{w})\tilde{w}}{\delta}\right] + [-sign(\tilde{w})\delta + \tilde{w}]^2\frac{sign(\tilde{w})\tilde{w}}{\delta}$$

$$\leq 2\tilde{w}^2 + 2\delta\,|\tilde{w}|$$

$$\leq 2\left[\eta_t\nabla F_k(\mathbf{w}_t^k, \xi_t^k)\right]_p^2 + 2\delta\left|\left[\eta_t\nabla F_k(\mathbf{w}_t^k, \xi_t^k)\right]_p\right|.$$

The expectation of $\left\|Q(\mathbf{w}_t^k - \eta_t\nabla F_k(\mathbf{w}_t^k, \xi_t^k)) - (\mathbf{w}_t^k - \eta_t\nabla F_k(\mathbf{w}_t^k, \xi_t^k))\right\|_2^2$ is:

$$\mathbb{E}\left\|Q(\mathbf{w}_t^k - \eta_t\nabla F_k(\mathbf{w}_t^k, \xi_t^k)) - (\mathbf{w}_t^k - \eta_t\nabla F_k(\mathbf{w}_t^k, \xi_t^k))\right\|_2^2$$

$$= \mathbb{E}(\mathbb{E}\left\|Q(\mathbf{w}_t^k - \eta_t\nabla F_k(\mathbf{w}_t^k, \xi_t^k)) - (\mathbf{w}_t^k - \eta_t\nabla F_k(\mathbf{w}_t^k, \xi_t^k))\right\|_2^2)$$

$$\leq 2\eta_t\delta\mathbb{E}\left\|\nabla F_k(\mathbf{w}_t^k, \xi_t^k))\right\|_1 + 2\eta_t^2\mathbb{E}\left\|\nabla F_k(\mathbf{w}_t^k, \xi_t^k))\right\|_2^2$$

$$= 2\delta\mathbb{E}\left\|\eta_t\nabla F_k(\mathbf{w}_t^k, \xi_t^k))\right\|_1 + 2\eta_t^2\mathbb{E}\left\|\nabla F_k(\mathbf{w}_t^k, \xi_t^k))\right\|_2^2$$

$$\leq 2\sqrt{d}\delta\eta_t^2\mathbb{E}\left\|\nabla F_k(\mathbf{w}_t^k, \xi_t^k))\right\|_2^2 + 2\eta_t^2\mathbb{E}\left\|\nabla F_k(\mathbf{w}_t^k, \xi_t^k))\right\|_2^2$$

$$= 2(\sqrt{d}\delta + 1)\eta_t^2\mathbb{E}\left\|\nabla F_k(\mathbf{w}_t^k, \xi_t^k))\right\|_2^2.$$

In the above inequality, we use the inequality $\|\cdot\|_1 \leq \sqrt{d}\|\cdot\|_2^2$, $d$ is the dimension of the vector. According to assumption 2 we can get:

$$\mathbb{E}\|\bar{\mathbf{w}}_{t+1} - \mathbf{w}^*\|_2^2 \leq 2(\sqrt{d}\delta + 1)\eta_t^2 G^2 + \mathbb{E}\|\bar{\mathbf{w}}_t - \eta_t\mathbf{g}_t - \mathbf{w}^*\|_2^2.$$

We next aim to bound $A_1$:

$$\|\bar{\mathbf{w}}_t - \eta_t\mathbf{g}_t - \mathbf{w}^*\|_2^2$$

$$= \|\bar{\mathbf{w}}_t - \eta_t \mathbf{g}_t - \mathbf{w}^* - \eta_t \bar{\mathbf{g}}_t + \eta_t \bar{\mathbf{g}}_t\|_2^2$$
$$= \underbrace{\|\bar{\mathbf{w}}_t - \eta_t \bar{\mathbf{g}}_t - \mathbf{w}^*\|_2^2}_{B_1} + \underbrace{\eta_t{}^2 \|\mathbf{g}_t - \bar{\mathbf{g}}_t\|_2^2}_{B_2} + \underbrace{2\,\eta_t \langle \bar{\mathbf{w}}_t - \eta_t \bar{\mathbf{g}}_t - \mathbf{w}^*, \mathbf{g}_t - \bar{\mathbf{g}}_t \rangle}_{B_3}.$$

Obviously $\mathbb{E}B_3 = 0$, so we can get:

$$\mathbb{E}\|\bar{\mathbf{w}}_t - \eta_t \mathbf{g}_t - \mathbf{w}^*\|_2^2 = \mathbb{E}\|\bar{\mathbf{w}}_t - \eta_t \bar{\mathbf{g}}_t - \mathbf{w}^*\|_2^2 + \mathbb{E}\eta_t{}^2 \|\mathbf{g}_t - \bar{\mathbf{g}}_t\|_2^2.$$

We next aim to bound $B_1$:

$$\|\bar{\mathbf{w}}_t - \eta_t \bar{\mathbf{g}}_t - \mathbf{w}^*\|_2^2$$
$$= \|\bar{\mathbf{w}}_t - \mathbf{w}^*\|_2^2 + \eta_t{}^2 \|\bar{\mathbf{g}}_t\|_2^2 - 2\eta_t \langle \bar{\mathbf{w}}_t - \mathbf{w}^*, \bar{\mathbf{g}}_t \rangle$$
$$= \|\bar{\mathbf{w}}_t - \mathbf{w}^*\|_2^2 + \underbrace{\eta_t{}^2 \|\bar{\mathbf{g}}_t\|_2^2}_{C_1} \underbrace{- 2\eta_t \frac{1}{N} \sum_{k=1}^{N} \langle \bar{\mathbf{w}}_t - \mathbf{w}_t{}^k, \nabla F_k(\mathbf{w}_t^k) \rangle}_{C_2}$$
$$\underbrace{- 2\eta_t \frac{1}{N} \sum_{k=1}^{N} \langle \mathbf{w}_t{}^k - \mathbf{w}^*, \nabla F_k(\mathbf{w}_t^k) \rangle}_{C_3}.$$

To bound $C_1$, $C_2$, $C_3$, we need to derive several inequalities using the properties of $F_1, F_2, \cdots, F_N$. First of all, for $k = 1, 2, \cdots, N$, $F_k$ is a $L$-smooth function, we can get:

$$F_k{}^* \le F_k(x) \le F_k(\mathbf{w}_t^k) + \langle \nabla F_k(\mathbf{w}_t^k), x - \mathbf{w}_t^k \rangle + \frac{L}{2}\|x - \mathbf{w}_t^k\|_2^2.$$

Specifically, when $x = \mathbf{w}_t^k - a\nabla F_k(\mathbf{w}_t^k)$, we can get:

$$F_k{}^* \le F_k(\mathbf{w}_t^k) + (\frac{L}{2}a^2 - a)\|\nabla F_k(\mathbf{w}_t^k) - \nabla F_k^*\|_2^2.$$

when $a = \frac{1}{L}$, we can get:

$$\left\|\nabla F_k(\mathbf{w}_t^k) - \nabla F_k{}^*\right\|_2^2 \le 2L(F_k(\mathbf{w}_t^k) - F_k{}^*). \tag{16}$$

According to the above equation, we can get:

$$\eta_t{}^2 \|\bar{\mathbf{g}}_t\|_2^2 = \eta_t{}^2 \left\|\sum_{k=1}^{N} p_k \nabla F_k(\mathbf{w}_t^k)\right\|_2^2 \le \eta_t{}^2 \sum_{k=1}^{N} p_k \|\nabla F_k(\mathbf{w}_t^k)\|_2^2 \le 2L\eta_t{}^2 \sum_{k=1}^{N} p_k(F_k(\mathbf{w}_t^k) - F_k{}^*). \tag{17}$$

In the inequality above, we have utilized inequality (16) and the property that $\|\cdot\|_2^2$ is a convex function.

For $k = 1, 2, \cdots, N$, $F_k$ is a $\mu$-strong function, so we can get:

$$-\langle \mathbf{w}_t^k - \mathbf{w}^*, \nabla F_k(\mathbf{w}_t^k) \rangle \le -(F_k(\mathbf{w}_t^k) - F_k(\mathbf{w}^*)) - \frac{\mu}{2}\|\mathbf{w}_t^k - \mathbf{w}^*\|_2^2. \tag{18}$$

By the inequality:

$$2\langle a, b \rangle \le \gamma\|a\|_2^2 + \gamma^{-1}\|b\|_2^2, (\gamma > 0). \tag{19}$$

we can get:

$$-2\langle \bar{\mathbf{w}}_t - \mathbf{w}_t^k, \nabla F_k(\mathbf{w}_t^k) \rangle \le \frac{1}{\eta_t}\|\bar{\mathbf{w}}_t - \mathbf{w}_t^k\|_2^2 + \eta_t\|\nabla F_k(\mathbf{w}_t^k)\|_2^2 \tag{20}$$

$$\le \frac{1}{\eta_t}\|\bar{\mathbf{w}}_t - \mathbf{w}_t^k\|_2^2 + 2L(F_k(\mathbf{w}_t^k) - F_k^*). \tag{21}$$

In the inequality above, we have utilized inequality (16) again. Now, we can use inequality (16), (17), (18), (20) to bound $C_1$, $C_2$, $C_3$ to bound $B_1$:

$$\|\bar{\mathbf{w}}_t - \eta_t \bar{\mathbf{g}}_t - \mathbf{w}^*\|_2^2$$

$$= \|\bar{\mathbf{w}}_t - \mathbf{w}^*\|_2^2 + {\eta_t}^2 \|\bar{\mathbf{g}}_t\|_2^2 - 2\eta_t \langle \bar{\mathbf{w}}_t - \mathbf{w}^*, \bar{\mathbf{g}}_t \rangle$$

$$= \|\bar{\mathbf{w}}_t - \mathbf{w}^*\|_2^2 + {\eta_t}^2 \|\bar{\mathbf{g}}_t\|_2^2 - 2\eta_t \frac{1}{N} \sum_{k=1}^{N} \langle \bar{\mathbf{w}}_t - \mathbf{w}_t{}^k + \mathbf{w}_t{}^k - \mathbf{w}^*, \nabla F_k(\mathbf{w}_t^k) \rangle$$

$$\leq \|\bar{\mathbf{w}}_t - \mathbf{w}^*\|_2^2 + 2L{\eta_t}^2 \sum_{k=1}^{N} p_k(F_k(\mathbf{w}_t^k) - F_k{}^*)$$

$$- 2\eta_t \frac{1}{N} \sum_{k=1}^{N} \langle \bar{\mathbf{w}}_t - \mathbf{w}_t{}^k, \nabla F_k(\mathbf{w}_t^k) \rangle + 2\eta_t \sum_{k=1}^{N} -\frac{1}{N} \langle \mathbf{w}_t{}^k - \mathbf{w}^*, \nabla F_k(\mathbf{w}_t^k) \rangle$$

$$= \|\bar{\mathbf{w}}_t - \mathbf{w}^*\|_2^2 + 4L{\eta_t}^2 \sum_{k=1}^{N} p_k(F_k(\mathbf{w}_t^k) - F_k{}^*) + \frac{1}{N} \sum_{k=1}^{N} \|\bar{\mathbf{w}}_t - \mathbf{w}_t^k\|_2^2$$

$$- 2\eta_t \frac{1}{N} \sum_{k=1}^{N} (F_k(\mathbf{w}_t^k) - F_k(\mathbf{w}^*)) - \eta_t \mu \frac{1}{N} \sum_{k=1}^{N} \|\mathbf{w}_t^k - \mathbf{w}^*\|_2^2$$

$$\leq \|\bar{\mathbf{w}}_t - \mathbf{w}^*\|_2^2 + 4L{\eta_t}^2 \sum_{k=1}^{N} p_k(F_k(\mathbf{w}_t^k) - F_k{}^*) + \frac{1}{N} \sum_{k=1}^{N} \|\bar{\mathbf{w}}_t - \mathbf{w}_t^k\|_2^2$$

$$- 2\eta_t \frac{1}{N} \sum_{k=1}^{N} (F_k(\mathbf{w}_t^k) - F_k(\mathbf{w}^*)) - \eta_t \mu \left\| \frac{1}{N} \sum_{k=1}^{N} (\mathbf{w}_t^k - \mathbf{w}^*) \right\|_2^2$$

$$= (1 - \eta_t \mu)\|\bar{\mathbf{w}}_t - \mathbf{w}^*\|_2^2 + \frac{1}{N} \sum_{k=1}^{N} \|\bar{\mathbf{w}}_t - \mathbf{w}_t^k\|_2^2$$

$$+ \underbrace{4L{\eta_t}^2 \sum_{k=1}^{N} p_k(F_k(\mathbf{w}_t^k) - F_k{}^*) - 2\eta_t \frac{1}{N} \sum_{k=1}^{N} (F_k(\mathbf{w}_t^k) - F_k(\mathbf{w}^*))}_{D}.$$

We set $\gamma_t = 2\eta_t(1 - 2L\eta_t)$, because $\eta_t \leq \frac{1}{4L}$, so $\eta_t \leq \gamma_t \leq 2\eta_t$. Decompose $D$ we can get:

$$D = 4L{\eta_t}^2 \sum_{k=1}^{N} p_k(F_k(\mathbf{w}_t^k) - F_k{}^*) - 2\eta_t \frac{1}{N} \sum_{k=1}^{N} (F_k(\mathbf{w}_t^k) - F_k(\mathbf{w}^*))$$

$$= (4L{\eta_t}^2 - 2\eta_t) \sum_{k=1}^{N} p_k F_k(\mathbf{w}_t^k) + 4L{\eta_t}^2 \sum_{k=1}^{N} p_k F_k{}^* + 2\eta_t \frac{1}{N} \sum_{k=1}^{N} F_k(\mathbf{w}^*)$$

$$= -\gamma_t \sum_{k=1}^{N} p_k F_k(\mathbf{w}_t^k) + 4L{\eta_t}^2 \sum_{k=1}^{N} p_k F_k{}^* + 2\eta_t F^*$$

$$= -\gamma_t \sum_{k=1}^{N} p_k(F_k(\mathbf{w}_t^k) - F^*) + (2\eta_t - \gamma_t) \sum_{k=1}^{N} p_k(F^* - F_k{}^*)$$

$$= -\gamma_t \sum_{k=1}^{N} p_k(F_k(\mathbf{w}_t^k) - F^*) + 4L{\eta_t}^2 \Gamma.$$

It should be noted that $\Gamma = \sum\limits_{k=1}^{N} p_k(F^* - F_k{}^*) = F^* - \sum\limits_{k=1}^{N} \frac{1}{N} F_k{}^*$, and it measures the degree of data heterogeneity between different client devices. We can bound the first term in $D$:

$$\sum_{k=1}^{N} p_k(F_k(\mathbf{w}_t^k) - F^*)$$

$$= \sum_{k=1}^{N} p_k(F_k(\mathbf{w}_t^k) - F_k(\bar{\mathbf{w}}_t)) + \sum_{k=1}^{N} p_k(F_k(\bar{\mathbf{w}}_t) - F^*)$$

$$\geq \sum_{k=1}^{N} p_k \left\langle \nabla F_k(\bar{\mathbf{w}}_t), \mathbf{w}_t^k - \bar{\mathbf{w}}_t \right\rangle + F(\bar{\mathbf{w}}_t) - F^*$$

$$\geq -\frac{1}{2} \sum_{k=1}^{N} p_k [\eta_t \|\nabla F_k(\bar{\mathbf{w}}_t)\|_2^2 + \frac{1}{\eta_t} \|\mathbf{w}_t^k - \bar{\mathbf{w}}_t\|_2^2] + F(\bar{\mathbf{w}}_t) - F^*$$

$$\geq -\sum_{k=1}^{N} p_k [\eta_t L(F_k(\bar{\mathbf{w}}_t) - F_k^*) + \frac{1}{2\eta_t} \|\mathbf{w}_t^k - \bar{\mathbf{w}}_t\|_2^2] + F(\bar{\mathbf{w}}_t) - F^*.$$

In the above inequation, we use the inequations (16),(19) and the convexity of $F_k(\cdot)$.

So we can bound $D$ and get:

$$D = -\gamma_t \sum_{k=1}^{N} p_k (F_k(\mathbf{w}_t^k) - F^*) + 4L\eta_t^2 \Gamma$$

$$\leq \gamma_t \sum_{k=1}^{N} p_k [\eta_t L(F_k(\bar{\mathbf{w}}_t) - F_k^*) + \frac{1}{2\eta_t} \|\mathbf{w}_t^k - \bar{\mathbf{w}}_t\|_2^2] - \gamma_t (F(\bar{\mathbf{w}}_t) - F^*) + 4L\eta_t^2 \Gamma$$

$$= \gamma_t \eta_t L \sum_{k=1}^{N} p_k (F_k(\bar{\mathbf{w}}_t) - F_k^*) + \frac{\gamma_t}{2\eta_t} \sum_{k=1}^{N} p_k \|\mathbf{w}_t^k - \bar{\mathbf{w}}_t\|_2^2 - \gamma_t (F(\bar{\mathbf{w}}_t) - F^*) + 4L\eta_t^2 \Gamma$$

$$= \gamma_t \eta_t L \sum_{k=1}^{N} p_k (F_k(\bar{\mathbf{w}}_t) - F^* + F^* - F_k^*) + \frac{\gamma_t}{2\eta_t} \sum_{k=1}^{N} p_k \|\mathbf{w}_t^k - \bar{\mathbf{w}}_t\|_2^2$$

$$- \gamma_t (\sum_{k=1}^{N} p_k F_k(\bar{\mathbf{w}}_t) - F^*) + 4L\eta_t^2 \Gamma$$

$$= \gamma_t (\eta_t L - 1) \sum_{k=1}^{N} p_k (F_k(\bar{\mathbf{w}}_t) - F^*) + (\gamma_t \eta_t L + 4L\eta_t^2) \Gamma + \frac{\gamma_t}{2\eta_t} \sum_{k=1}^{N} p_k \|\mathbf{w}_t^k - \bar{\mathbf{w}}_t\|_2^2$$

$$\leq 6L\eta_t^2 \Gamma + \sum_{k=1}^{N} p_k \|\mathbf{w}_t^k - \bar{\mathbf{w}}_t\|_2^2.$$

When $t + 1 \notin \mathcal{I}_E$, we can conclude that:

$$\mathbb{E} \|\bar{\mathbf{w}}_{t+1} - \mathbf{w}^*\|_2^2 \leq 2(\sqrt{d}\delta + 1)\eta_t^2 G^2 + (1 - \eta_t \mu)\mathbb{E} \|\bar{\mathbf{w}}_t - \mathbf{w}^*\|_2^2$$

$$+ \frac{2}{N} \sum_{K=1}^{N} \|\bar{\mathbf{w}}_t - \mathbf{w}_t^k\|_2^2 + 6L\eta_t^2 \Gamma + \mathbb{E}\eta_t^2 \|\mathbf{g}_t - \bar{\mathbf{g}}_t\|_2^2.$$

When $t + 1 \in \mathcal{I}_E$, we have $\mathbf{w}_{t+1}^k = Q(\frac{\sum_{k=1}^{N} \mathbf{v}_{t+1}^k}{N})$ :

$$\mathbb{E} \|\bar{\mathbf{w}}_{t+1} - \mathbf{w}^*\|_2^2$$

$$= \mathbb{E} \left\| Q(\frac{1}{N} \sum_{k=1}^{N} \mathbf{v}_{t+1}^k) - \mathbf{w}^* \right\|_2^2$$

$$= \mathbb{E} \left\| Q(\frac{1}{N} \sum_{k=1}^{N} \mathbf{v}_{t+1}^k) - \frac{1}{N} \sum_{k=1}^{N} \mathbf{v}_{t+1}^k + \frac{1}{N} \sum_{k=1}^{N} \mathbf{v}_{t+1}^k - \mathbf{w}^* \right\|_2^2$$

$$= \mathbb{E} \left\| Q(\frac{1}{N} \sum_{k=1}^{N} \mathbf{v}_{t+1}^k) - \frac{1}{N} \sum_{k=1}^{N} \mathbf{v}_{t+1}^k + \frac{1}{N} \sum_{k=1}^{N} Q(\mathbf{w}_t^k - \eta_t \nabla F_k(\mathbf{w}_t^k, \xi_t^k)) - \mathbf{w}^* \right\|_2^2$$

$$\leq \underbrace{\mathbb{E} \left\| Q(\frac{1}{N} \sum_{k=1}^{N} \mathbf{v}_{t+1}^k) - \frac{1}{N} \sum_{k=1}^{N} \mathbf{v}_{t+1}^k \right\|_2^2}_{E_1} + \underbrace{\mathbb{E} \left\| \frac{1}{N} \sum_{k=1}^{N} Q(\mathbf{w}_t^k - \eta_t \nabla F_k(\mathbf{w}_t^k, \xi_t^k)) - \mathbf{w}^* \right\|_2^2}_{E_2}$$

$$+ 2\mathbb{E}\langle Q(\frac{1}{N}\sum_{k=1}^{N}\mathbf{v}_{t+1}^{k}) - \frac{1}{N}\sum_{k=1}^{N}\mathbf{v}_{t+1}^{k}, \underbrace{\frac{1}{N}\sum_{k=1}^{N}Q(\mathbf{w}_{t}^{k} - \eta_{t}\nabla F_{k}(\mathbf{w}_{t}^{k},\xi_{t}^{k})) - \mathbf{w}^{*}\rangle}_{E_3}$$

$E_2$ has been given constraints in the text above that:

$$E_2 \leq 2(\sqrt{d}\delta + 1)\eta_t^2 G^2 + (1 - \eta_t\mu)\mathbb{E}\left\|\bar{\mathbf{w}}_t - \mathbf{w}^*\right\|_2^2$$
$$+ \frac{2}{N}\sum_{K=1}^{N}\left\|\bar{\mathbf{w}}_t - \mathbf{w}_t^k\right\|_2^2 + 6L\eta_t^2\Gamma + \mathbb{E}\eta_t^2\left\|\mathbf{g}_t - \bar{\mathbf{g}}_t\right\|_2^2.$$

According to the law of iterated expectations, $E_3$ is 0. We next constrain $E_1$. According to our assumption before Lemma 2, we have:

$$E_1 = \mathbb{E}E_1 \leq \mathbb{E}d\delta^2 \leq \mathbb{E}d\eta_t^2\left\|\nabla F_k(\mathbf{w}_t^k,\xi_t^k)\right\|_2^2 = d\eta_t^2 G^2.$$

So when $t + 1 \in \mathcal{I}_E$ we have:

$$\mathbb{E}\left\|\bar{\mathbf{w}}_{t+1} - \mathbf{w}^*\right\|_2^2 \leq 2(\sqrt{d}\delta + 1 + \frac{d}{2})\eta_t^2 G^2 + (1 - \eta_t\mu)\mathbb{E}\left\|\bar{\mathbf{w}}_t - \mathbf{w}^*\right\|_2^2 + 6L\eta_t^2\Gamma$$
$$+ \frac{2}{N}\sum_{K=1}^{N}\left\|\bar{\mathbf{w}}_t - \mathbf{w}_t^k\right\|_2^2 + \mathbb{E}\eta_t^2\left\|\mathbf{g}_t - \bar{\mathbf{g}}_t\right\|_2^2.$$

In summary, whether $t + 1 \in \mathcal{I}_E$ or $t + 1 \notin \mathcal{I}_E$, we have:

$$\mathbb{E}\left\|\bar{\mathbf{w}}_{t+1} - \mathbf{w}^*\right\|_2^2 \leq 2(\sqrt{d}\delta + 1 + \frac{d}{2})\eta_t^2 G^2 + (1 - \eta_t\mu)\mathbb{E}\left\|\bar{\mathbf{w}}_t - \mathbf{w}^*\right\|_2^2 + 6L\eta_t^2\Gamma$$
$$+ \frac{2}{N}\sum_{K=1}^{N}\left\|\bar{\mathbf{w}}_t - \mathbf{w}_t^k\right\|_2^2 + \mathbb{E}\eta_t^2\left\|\mathbf{g}_t - \bar{\mathbf{g}}_t\right\|_2^2$$

$\square$

**Lemma 3.** *Based on assumption 2, we can get:*

$$\mathbb{E}\|\mathbf{g}_t - \bar{\mathbf{g}}_t\|_2^2 \leq \frac{1}{N^2}\sum_{k=1}^{N}\sigma_k{}^2.$$

*Proof.*

$$\mathbb{E}\|\mathbf{g}_t - \bar{\mathbf{g}}_t\|_2^2 = \mathbb{E}\left\|\frac{1}{N}\sum_{k=1}^{N}(\nabla F_k(\mathbf{w}_t^k,\xi_t^k) - \nabla F_k(\mathbf{w}_t^k))\right\|_2^2$$
$$= \sum_{k=1}^{N}\frac{1}{N^2}\mathbb{E}\left\|\nabla F_k(\mathbf{w}_t^k,\xi_t^k) - \nabla F_k(\mathbf{w}_t^k)\right\|_2^2$$
$$\leq \frac{1}{N^2}\sum_{k=1}^{N}\sigma_k{}^2.$$

$\square$

**Lemma 4.** *If $\eta_t$ is non-increasing and $\eta_t \leq 2\eta_{t+E}, (t \geq 0)$, based on assumption 2, we can get:*

$$\frac{1}{N}\sum_{k=1}^{N}\mathbb{E}\left\|\bar{\mathbf{w}}_t - \mathbf{w}_t^k\right\|_2^2 \leq 8E^2\eta_t^2 G^2(2\sqrt{d}\delta + 3).$$

*Proof.* Since FedAvg requires a communication each $E$ steps. Therefore, for any $t \geq 0$, there exists a $t_0 \leq t$, such that $t - t_0 \leq E - 1$ and $\bar{\mathbf{w}}_{t_0} = \mathbf{w}_{t_0}^k$ for all k. Also, we use the fact that $\eta_t$ is non-increasing and $\eta_t \leq 2\eta_{t+E}, (t \geq 0)$, then:

$$\frac{1}{N}\sum_{k=1}^{N}\mathbb{E}\big\|\bar{\mathbf{w}}_t - \mathbf{w}_t^k\big\|_2^2$$

$$= \frac{1}{N}\mathbb{E}\sum_{k=1}^{N}\big\|\mathbf{w}_t^k - \bar{\mathbf{w}}_{t_0} - (\bar{\mathbf{w}}_t - \bar{\mathbf{w}}_{t_0})\big\|_2^2$$

$$\leq \frac{1}{N}\sum_{k=1}^{N}\mathbb{E}\big\|\mathbf{w}_t^k - \bar{\mathbf{w}}_{t_0}\big\|_2^2$$

$$= \frac{1}{N}\sum_{k=1}^{N}\mathbb{E}\big\|\mathbf{w}_t^k - \mathbf{w}_{t_0}\big\|_2^2$$

$$= \frac{1}{N}\sum_{k=1}^{N}\mathbb{E}\big\|\mathbf{w}_t^k - \mathbf{w}_{t-1}^k + \mathbf{w}_{t-1}^k - \mathbf{w}_{t-2}^k + \ldots + \mathbf{w}_{t_0+1}^k - \mathbf{w}_{t_0}^k\big\|_2^2$$

$$\leq \frac{1}{N}\sum_{k=1}^{N}E\sum_{h=t_0}^{t-1}\mathbb{E}\big\|\mathbf{w}_{h+1}^k - \mathbf{w}_h^k\big\|_2^2$$

$$= \frac{1}{N}\sum_{k=1}^{N}E\sum_{h=t_0}^{t-1}\mathbb{E}\big\|Q(\mathbf{w}_h^k - \eta_h\nabla F_h(\mathbf{w}_h^k, \xi_h^k)) - (\mathbf{w}_h^k - \eta_h\nabla F_h(\mathbf{w}_h^k, \xi_h^k)) - \eta_h\nabla F_h(\mathbf{w}_h^k, \xi_h^k)\big\|_2^2$$

$$\leq \frac{1}{N}\sum_{k=1}^{N}2E^2(\mathbb{E}\big\|Q(\mathbf{w}_h^k - \eta_h\nabla F_h(\mathbf{w}_h^k, \xi_h^k)) - (\mathbf{w}_h^k - \eta_h\nabla F_h(\mathbf{w}_h^k, \xi_h^k))\big\|_2^2$$

$$+ \mathbb{E}\big\|\eta_h\nabla F_h(\mathbf{w}_h^k, \xi_h^k)\big\|_2^2).$$

Inequalities $\left\|\sum_{i=1}^{E} a_i\right\|_2^2 \leq E\sum_{i=1}^{E}\|a_i\|_2^2$ and $\mathbb{E}\left\|X - \mathbb{E}X\right\|_2^2 \leq \mathbb{E}\left\|X\right\|_2^2$ are used in the above proof.

We have already proofed in lemma 2 that:

$$\mathbb{E}\big\|Q(\mathbf{w}_h^k - \eta_h\nabla F_h(\mathbf{w}_h^k, \xi_h^k)) - (\mathbf{w}_h^k - \eta_h\nabla F_h(\mathbf{w}_h^k, \xi_h^k))\big\|_2^2 \leq 2(\sqrt{d}\delta + 1)\eta_h^2 G^2.$$

So we can have:

$$\frac{1}{N}\sum_{k=1}^{N}\mathbb{E}\big\|\bar{\mathbf{w}}_t - \mathbf{w}_t^k\big\|_2^2 \leq 8E^2\eta_t^2 G^2(2\sqrt{d}\delta + 3).$$

$\square$

**Theorem 1.** *Under the condition that assumptions 1, 2, we choose $\kappa = \frac{L}{\mu}$ , $\gamma = \max\{8\kappa, E\} - 1$ and $\eta_t = \frac{\mu}{2(t+\gamma)}$. When $t$ satisfying $\delta^2 \leq \eta_t^2 G^2$ our low precision federated learning algorihm with full device participation satisfies:*

$$\mathbb{E}F(\bar{\mathbf{w}}_t) - F^* \leq \frac{L}{2}\frac{v}{t+\gamma} \leq \frac{\kappa}{t+\gamma}\big(\frac{2B}{\mu} + \frac{\mu(\gamma+1)}{2}\|\mathbf{w}_1 - \mathbf{w}^*\|^2\big).$$

*where*

$$B = 2(\sqrt{d}\delta + 1 + \frac{d}{2})G^2 + 16E^2G^2(2\sqrt{d}\delta + 3) + \frac{1}{N^2}\sum_{k=1}^{N}\sigma_k^2 + 6L\Gamma.$$

*Proof.* Set $\Delta_t = \mathbb{E}\|\bar{\mathbf{w}}_t - \mathbf{w}^*\|_2^2$, according to lemma 1, 2, 3, we can have:

$$\Delta_{t+1} \leq (1 - \mu\eta_t)\Delta_t + \eta_t^2 B.$$

Set $\eta_t = \frac{\beta}{t+\gamma}$, we can choose appropriate $\beta$ and $\gamma$ so that $\eta_t \leq \min\{\frac{1}{\mu}, \frac{1}{4L}\}$ and $\eta_t \leq 2\eta_{t+E}$. Set $v = \max\{\frac{\beta^2 B}{\beta\mu - 1}, (\gamma+1)\Delta_1\}$, we will prove $\Delta_t \leq \frac{v}{t+\gamma}$ by mathematical induction.

When $t = 1$, obviously it holds. Assume the conclusion holds for some $t$, it follows that:

$$\Delta_{t+1} \leq (1 - \mu\eta_t)\Delta_t + \eta_t^2 B$$
$$\leq (1 - \frac{\beta\mu}{t+\gamma})\frac{v}{t+\gamma} + \frac{\beta^2 B}{(t+\gamma)^2}$$
$$= \frac{t+\gamma - 1}{(t+\gamma)^2}v + [\frac{\beta^2 B}{(t+\gamma)^2} - \frac{\beta\mu - 1}{(t+\gamma)^2}v]$$
$$\leq \frac{v}{t+\gamma + 1}.$$

Then by the L-smoothness of $F(\cdot)$, we have:

$$\mathbb{E}F(\bar{\mathbf{w}}_t) - F^* \leq \frac{L}{2}\Delta_t \leq \frac{L}{2}\frac{v}{t+\gamma}.$$

We choose $\beta = \frac{2}{\mu}, \gamma = \max\{8\kappa, E\} - 1$ and denote $\kappa = \frac{L}{\mu}$, then we have:

$$v = \max\{\frac{\beta^2 B}{\beta\mu - 1}, (\gamma+1)\Delta_1\} \leq \frac{\beta^2 B}{\beta\mu - 1} + (\gamma+1)\Delta_1 \leq \frac{4B}{\mu^2} + (\gamma+1)\Delta_1.$$

and

$$\mathbb{E}F(\bar{\mathbf{w}}_t) - F^* \leq \frac{L}{2}\frac{v}{t+\gamma} \leq \frac{\kappa}{t+\gamma}(\frac{2B}{\mu} + \frac{\mu(\gamma+1)}{2}\Delta_1).$$

$\square$

# B  Proofs of Theorem 2

We analyze our low precision federated learning algorithm in the setting of partial device participation in this section.

**Updating scheme** In real world application scenarios, constrained by communication efficiency and low straggler effect, FedAvg initiates by selecting a random subset $\mathcal{S}_t$ which length is set to $K$, and subsequently carries out updates exclusively on this subset. The analysis becomes somewhat complex due to the variability of $\mathcal{S}_t$ every $E$ steps. Nevertheless, we can employ a thought trick to address this challenge. We envision that FedAvg initiates each epoch by engaging **all devices** and then relies solely on the parameters updated on a subset of these devices to generate the parameters for the subsequent round. It is evident that this method of parameter update is equivalent to the original approach. Then the update of FedAvg with partial devices active can be described as: for all $k \in [N]$,

$$\mathbf{v}_{t+1}^k = Q(w_t^k - \eta_t \nabla F_k(\mathbf{w}_t^k, \xi_t^k)).$$

$$\mathbf{w}_{t+1}^k = \begin{cases} \mathbf{v}_{t+1}^k & \text{if } t+1 \notin \mathcal{S}_t, \\ Q(\sum_{k \in \mathcal{S}_t} \frac{1}{K}\mathbf{v}_{t+1}^k) & \text{if } t+1 \in \mathcal{S}_t. \end{cases}$$

We use $\mathbb{E}_{\mathcal{S}_t}(\cdot)$ to eliminate the error caused by $\mathcal{S}_t$. Unless otherwise specified, the meanings and assumptions of the symbols used in this section are the same as in the previous section.

**Lemma 5.** *If $t+1 \in \mathcal{I}_E$, we have:*

$$\mathbb{E}_{\mathcal{S}_t}(\bar{\mathbf{w}}_{t+1}) = \bar{\mathbf{v}}_{t+1}.$$

*Proof.* According to the selection method of $\mathcal{S}_t$, there is a function:

$$\mathbb{I}_i = \begin{cases} 1, & \mathbb{P}(i \in \mathcal{S}_t) = \frac{K}{N} \\ 0, & \mathbb{P}(i \notin \mathcal{S}_t) = \frac{N-K}{N}. \end{cases}$$

According to the aggregation process, we have:

$$\bar{\mathbf{w}}_{t+1} = Q\left(\frac{1}{K}\sum_{i=1}^{N}\mathbb{I}_i\mathbf{v}_{t+1}^i\right).$$

So we have:

$$\mathbb{E}_{\mathcal{S}_t}(\bar{\mathbf{w}}_{t+1}) = \mathbb{E}_{\mathcal{S}_t}\left(\frac{1}{K}\sum_{i=1}^{N}\mathbb{I}_i\mathbf{v}_{t+1}^i\right)$$

$$= \frac{1}{K}\sum_{i=1}^{N}\mathbb{P}_i\mathbf{v}_{t+1}^i = \frac{1}{K}\sum_{i=1}^{N}\frac{K}{N}\mathbf{v}_{t+1}^i = \sum_{i=1}^{N}\frac{1}{N}\mathbf{v}_{t+1}^i = \bar{\mathbf{v}}_{t+1}.$$

$\square$

**Lemma 6.** *For $t + 1 \in \mathcal{I}_E$ we have:*

$$\mathbb{E}_{\mathcal{S}_t}\|\bar{\mathbf{v}}_{t+1} - \bar{\mathbf{w}}_{t+1}\|_2^2 \le \frac{N-K}{N-1}\frac{8}{K}E^2\eta_t^2G^2(2\sqrt{d}\delta+3) + dG^2\eta_t^2.$$

*Proof.* According to the aggregation process, we have $\bar{\mathbf{w}}_{t+1} = Q\left(\frac{1}{K}\sum_{i\in\mathcal{S}_t}\mathbf{v}_{t+1}^i\right).$

$$\mathbb{E}_{\mathcal{S}_t}\|\bar{\mathbf{w}}_{t+1} - \bar{\mathbf{v}}_{t+1}\|_2^2$$

$$= \mathbb{E}_{\mathcal{S}_t}\left\|Q\left(\frac{1}{K}\sum_{i\in\mathcal{S}_t}\mathbf{v}_{t+1}^i\right) - \frac{1}{K}\sum_{i\in\mathcal{S}_t}\mathbf{v}_{t+1}^i + \frac{1}{K}\sum_{i\in\mathcal{S}_t}\mathbf{v}_{t+1}^i - \bar{\mathbf{v}}_{t+1}\right\|_2^2$$

$$= \underbrace{\mathbb{E}_{\mathcal{S}_t}\left\|Q\left(\frac{1}{K}\sum_{i\in\mathcal{S}_t}\mathbf{v}_{t+1}^i\right) - \frac{1}{K}\sum_{i\in\mathcal{S}_t}\mathbf{v}_{t+1}^i\right\|_2^2}_{F_1} + \underbrace{\frac{1}{K^2}\mathbb{E}_{\mathcal{S}_t}\left\|\sum_{i=1}^{N}\mathbb{I}(i\in\mathcal{S}_t)(\mathbf{v}_{t+1}^i - \bar{\mathbf{v}}_{t+1})\right\|_2^2}_{F_2}$$

$$+ \underbrace{2\left\langle Q\left(\frac{1}{K}\sum_{i\in\mathcal{S}_t}\mathbf{v}_{t+1}^i\right) - \frac{1}{K}\sum_{i\in\mathcal{S}_t}\mathbf{v}_{t+1}^i, \frac{1}{K}\sum_{i\in\mathcal{S}_t}\mathbf{v}_{t+1}^i - \bar{\mathbf{v}}_{t+1}\right\rangle}_{F_3}$$

In the lemma 2, we have proven $F_1 \le dG^2\eta_t^2$. And obviously we have $F_3 = 0$, we next aim to bound $F_2$.

$$\frac{1}{K^2}\mathbb{E}_{\mathcal{S}_t}\left\|\sum_{i=1}^{N}\mathbb{I}(i\in\mathcal{S}_t)(\mathbf{v}_{t+1}^i - \bar{\mathbf{v}}_{t+1})\right\|_2^2$$

$$= \frac{1}{K^2}\sum_{i=1}^{N}\mathbb{P}(i\in\mathcal{S}_t)\|\mathbf{v}_{t+1}^i - \bar{\mathbf{v}}_{t+1}\|_2^2$$

$$+ \frac{1}{K^2}\sum_{i\ne j}\mathbb{P}(i,j\in\mathcal{S}_t)\left\langle\mathbf{v}_{t+1}^i - \bar{\mathbf{v}}_{t+1}, \mathbf{v}_{t+1}^j - \bar{\mathbf{v}}_{t+1}\right\rangle$$

$$= \frac{1}{KN}\sum_{i=1}^{N}\|\mathbf{v}_{t+1}^i - \bar{\mathbf{v}}_{t+1}\|_2^2 + \sum_{i\ne j}\frac{K-1}{KN(N-1)}\langle\mathbf{v}_{t+1}^i - \bar{\mathbf{v}}_{t+1}, \mathbf{v}_{t+1}^j - \bar{\mathbf{v}}_{t+1}\rangle$$

$$= \frac{1}{K(N-1)}\left(1 - \frac{K}{N}\right)\sum_{i=1}^{N}\|\mathbf{v}_{t+1}^i - \bar{\mathbf{v}}_{t+1}\|_2^2.$$

The last step in the above equation uses:

$$\sum_{i=1}^{N}\|\mathbf{v}_{t+1}^i - \bar{\mathbf{v}}_{t+1}\|_2^2 + \sum_{i\ne j}\langle\mathbf{v}_{t+1}^i - \bar{\mathbf{v}}_{t+1}, \mathbf{v}_{t+1}^j - \bar{\mathbf{v}}_{t+1}\rangle = 0.$$

Therefore:

$$\mathbb{E}\left\|\bar{\mathbf{w}}_{t+1}-\bar{\mathbf{v}}_{t+1}\right\|_2^2=\frac{N}{K(N-1)}\left(1-\frac{K}{N}\right)\mathbb{E}\left[\frac{1}{N}\sum_{i=1}^N\left\|\mathbf{v}_{t+1}^i-\bar{\mathbf{w}}_{t_0}+\bar{\mathbf{w}}_{t_0}-\bar{\mathbf{v}}_{t+1}\right\|_2^2\right]+F_1$$

$$\leq\frac{N}{K(N-1)}\left(1-\frac{K}{N}\right)\mathbb{E}\left[\frac{1}{N}\sum_{i=1}^N\left\|\mathbf{v}_{t+1}^i-\bar{\mathbf{w}}_{t_0}\right\|_2^2\right]+F_1$$

$$\leq\frac{N}{K(N-1)}\left(1-\frac{K}{N}\right)8E^2\eta_t^2G^2(2\sqrt{d}\delta+3)+dG^2\eta_t^2.$$

We used $\mathbb{E}\left\|\mathbf{X}-\mathbb{E}\mathbf{X}\right\|_2^2\leq\mathbb{E}\left\|\mathbf{X}\right\|_2^2$ in the inequality above. $\qquad\square$

**Theorem 2.** *Under the condition that assumptions 1, 2, 3, we choose* $\kappa=\frac{L}{\mu}$, $\gamma=\max\{8\kappa,E\}-1$, $\eta_t=\frac{\mu}{2(t+\gamma)}$ *and* $B=2(\sqrt{d}\delta+1+\frac{d}{2})G^2+16E^2G^2(2\sqrt{d}\delta+3)+\frac{1}{N^2}\sum_{k=1}^K\sigma_k^2+6L\Gamma$, $C=\frac{N-K}{N-1}\frac{8}{K}E^2G^2(2\sqrt{d}\delta+3)+dG^2$. *When t satisfying* $\delta^2\leq\eta_t^2G^2$, *FedAvg with quantization and partial device participation satisfies:*

$$\mathbb{E}[F(\bar{\mathbf{w}}_t)]-F^*\leq\frac{\kappa}{\gamma+t}\left(\frac{2(B+C)}{\mu}+\frac{\mu(\gamma+1)}{2}\|\mathbf{w}_1-\mathbf{w}^*\|^2\right).$$

*Proof.* We have:

$$\mathbb{E}\left\|\bar{\mathbf{w}}_{t+1}-\mathbf{w}^*\right\|_2^2=\mathbb{E}\left\|\bar{\mathbf{w}}_{t+1}-\bar{\mathbf{v}}_{t+1}+\bar{\mathbf{v}}_{t+1}-\mathbf{w}^*\right\|_2^2$$

$$=\underbrace{\mathbb{E}\left\|\bar{\mathbf{w}}_{t+1}-\bar{\mathbf{v}}_{t+1}\right\|_2^2}_{G_1}+\underbrace{\mathbb{E}\left\|\bar{\mathbf{v}}_{t+1}-\mathbf{w}^*\right\|_2^2}_{G_2}+\underbrace{2\mathbb{E}\langle\bar{\mathbf{w}}_{t+1}-\bar{\mathbf{v}}_{t+1},\bar{\mathbf{v}}_{t+1}-\mathbf{w}^*\rangle}_{G_3}.$$

We take the expectation for the above formula, $G_3$ vanishes according to lemma 5.

If $t+1\notin\mathcal{I}_E$, according to our parameter update settings, $G_1$ vanishes. We use lemma 1, 2, 3, 4 to bound $G_2$:

$$\mathbb{E}\left\|\bar{\mathbf{v}}_{t+1}-\mathbf{w}^*\right\|_2^2=\mathbb{E}\left\|\bar{\mathbf{w}}_{t+1}-\mathbf{w}^*\right\|_2^2\leq(1-\eta_t\mu)\mathbb{E}\left\|\bar{\mathbf{w}}_t-\mathbf{w}^*\right\|_2^2+\eta_t^2B.$$

If $t+1\in\mathcal{I}_E$, We use lemma 6 to bound $G_1$, lemma 1, 2, 3, 4 to bound $G_2$, then we have :

$$\mathbb{E}\left\|\bar{\mathbf{w}}_{t+1}-\mathbf{w}^*\right\|_2^2=\mathbb{E}\left\|\bar{\mathbf{w}}_{t+1}-\bar{\mathbf{v}}_{t+1}\right\|_2^2+\mathbb{E}\left\|\bar{\mathbf{v}}_{t+1}-\mathbf{w}^*\right\|_2^2$$

$$\leq(1-\eta_t\mu)\mathbb{E}\left\|\bar{\mathbf{w}}_t-\mathbf{w}^\star\right\|_2^2+\eta_t^2(B+C).$$

We use exactly the same method as in Theorem 1 and then we have:

$$\mathbb{E}[F(\bar{\mathbf{w}}_t)]-F^*\leq\frac{\kappa}{\gamma+t}\left(\frac{2(B+C)}{\mu}+\frac{\mu(\gamma+1)}{2}\|w_1-w^*\|^2\right).$$

$\qquad\square$

## C   Hyperparameters Setting

We will introduce the four FL methods, ABAvg, FedProx, FedGen, FedFTG, and their hyperparameters setting in our experiments to prove our method's effectiveness.

- ABAvg [35] is a data-dependent distillation FL method, which uses a validation dataset $\mathcal{D}_v$ to reweight each collected devices. In our experiment, we set the test dataset as the validation dataset. For FashionMNIST, CIFAR10, CIFAR100, we use the whole test dataset, and for CINIC10, we choose the $20\%$ data of each label.

- FedProx [22] is an FL method with regularization when training locally. We set the FedProx proximal term $\mu=0.1$ in our experiment.

- FedGen [42] is a data-free distillation FL method, which maintain a generator in the server to generate hidden variables, and transfer the generator to the device side for data enhancement. We set the generator learning rate to $3e-4$, hidden dimension to 32. The generator's training epoch is 5.
- FedFTG [40] is a data-free distillation FL method, which maintain a generator on the server for fine-tuning aggregated server parameters. We set the generator learning rate to $1e-2$, the fine-tuning learning rate to $1e-4$. The totally fine-tune epoch is set to 10, inner epoch for generator is 1, inner epoch for server parameter is 5.

All of our models are trained on s GeForce RTX 4090. We found only very small differences when evaluating on these other hardware platforms.

## D  Low Precision Training with Other FL Methods

We put the four algorithms used here together with Efficient FL. Judging from the Table 4, low-precision local training is sufficient in FL system. We put more emphasis on showing the case of $\alpha = 0.01$, because this case is more consistent with the actual situation of FL, that is, the data heterogeneity of edge devices. Combining four FL methods and four datasets, in the case of $\alpha = 0.01$, Figure 4, we can find the fact that our algorithm can maintain or even surpass the accuracy of the original algorithm on 8 bit, and at the same time, it can do the same on 6 bit to maintain accuracy.

Table 4: Low precision FL on ABAvg, FedProx, FedGen, FedFTG. Our method is universal, whether it is regularized locally to constrain updates or finetuned on the server. Our method can maintain the effectiveness of various FL methods with a certain training cost, and can be applied in both regularization and distillation.

| Method | Avaging | Prec. | $\alpha = 0.01$ | | | | $\alpha = 0.04$ | | | | $\alpha = 0.16$ | | | |
|---|---|---|---|---|---|---|---|---|---|---|---|---|---|---|
| | | | FMNIST | CIFAR10 | CINIC10 | CIFAR100 | FMNIST | CIFAR10 | CINIC10 | CIFAR100 | FMNIST | CIFAR10 | CINIC10 | CIFAR100 |
| ABAvg | w avg | 8 bit | 80.7±0.4 | 51.1±1.4 | 41.4±0.8 | 22.4±0.9 | 88.4±1.2 | 55.8±0.5 | 50.9±2.7 | 34.3±0.2 | 90.2±0.6 | 72.3±1.0 | 58.2±1.4 | 42.5±0.3 |
| | | 6 bit | 78.0±0.7 | 46.8±0.6 | 39.1±1.3 | 14.4±0.2 | 86.1±0.5 | 53.6±0.6 | 42.6±0.5 | 22.0±0.7 | 89.9±0.4 | 70.2±1.3 | 47.3±0.3 | 33.2±0.1 |
| | | 32 bit | 79.5±1.1 | 50.2±1.0 | 40.4±2.0 | 20.3±0.7 | 88.4±0.8 | 56.5±0.9 | 49.9±2.1 | 31.4±0.3 | 90.8±0.5 | 72.7±1.1 | 58.4±0.9 | 41.7±0.4 |
| | w/o avg | 8 bit | 68.1±5.5 | 30.5±1.2 | 22.6±2.6 | 7.45±0.6 | 79.4±1.6 | 40.2±4.1 | 35.8±1.3 | 14.0±2.9 | 89.1±0.8 | 63.8±1.7 | 48.2±2.2 | 25.2±0.3 |
| | | 6 bit | 60.5±2.8 | 25.5±1.0 | 18.4±0.8 | 3.42±0.6 | 71.2±0.5 | 33.2±2.1 | 27.3±1.1 | 7.01±0.1 | 83.3±0.6 | 41.8±2.5 | 40.1±1.3 | 9.85±0.2 |
| | | 32 bit | 78.4±2.8 | 44.5±4.2 | 26.4±3.8 | 16.0±0.4 | 83.5±2.3 | 56.1±3.1 | 42.1±3.8 | 27.3±0.6 | 90.3±1.0 | 71.3±1.3 | 58.1±1.6 | 39.9±0.7 |
| FedProx | w avg | 8 bit | 83.0±0.6 | 51.4±0.5 | 42.2±1.4 | 18.2±0.1 | 86.5±1.2 | 63.2±0.3 | 48.4±2.4 | 35.5±0.5 | 88.8±0.7 | 71.9±1.0 | 57.6±0.6 | 43.7±0.1 |
| | | 6 bit | 78.1±0.4 | 48.3±1.0 | 37.9±0.2 | 17.4±0.4 | 79.9±0.2 | 52.0±0.4 | 38.1±0.1 | 21.3±0.2 | 82.8±0.2 | 56.2±0.2 | 44.6±0.2 | 25.3±0.1 |
| | | 32 bit | 82.9±0.9 | 52.6±0.5 | 42.4±1.0 | 18.8±0.1 | 86.9±0.6 | 64.9±0.9 | 49.2±1.9 | 36.2±0.6 | 89.3±0.6 | 71.3±0.5 | 57.2±1.3 | 43.9±0.1 |
| | w/o avg | 8 bit | 72.6±1.0 | 27.0±3.9 | 25.7±2.2 | 6.01±0.3 | 81.5±1.9 | 33.6±3.9 | 33.0±1.5 | 14.2±0.5 | 86.4±0.6 | 59.8±2.3 | 50.0±2.3 | 27.8±0.3 |
| | | 6 bit | 60.8±2.7 | 18.2±1.4 | 15.9±1.2 | 2.71±0.1 | 71.8±1.7 | 27.4±1.5 | 22.4±0.9 | 6.62±0.2 | 81.3±0.5 | 40.3±6.6 | 38.3±0.7 | 10.9±0.3 |
| | | 32 bit | 81.1±3.8 | 45.1±3.4 | 32.2±3.6 | 16.8±0.7 | 86.3±1.5 | 57.0±2.1 | 46.0±2.0 | 29.3±1.3 | 89.1±0.4 | 72.5±0.5 | 58.0±0.7 | 43.2±0.7 |
| FedGen | w avg | 8 bit | 79.9±2.2 | 45.5±1.9 | 34.7±2.1 | 20.1±1.1 | 86.2±0.7 | 60.5±1.4 | 47.4±1.3 | 32.9±0.7 | 91.1±0.3 | 74.0±0.9 | 58.4±1.0 | 43.2±0.2 |
| | | 6 bit | 78.5±1.3 | 44.8±2.0 | 30.1±0.4 | 15.4±0.2 | 81.9±1.3 | 41.4±0.3 | 34.0±0.2 | 9.20±0.1 | 85.8±0.7 | 62.6±0.7 | 55.3±0.7 | 14.0±0.1 |
| | | 32 bit | 78.9±1.5 | 47.6±2.2 | 35.3±2.6 | 16.1±0.8 | 83.6±1.2 | 56.9±2.9 | 48.2±1.3 | 32.7±1.0 | 91.0±0.6 | 74.2±0.6 | 58.8±0.5 | 42.6±0.4 |
| | w/o avg | 8 bit | 71.1±3.2 | 31.0±2.4 | 25.2±1.5 | 2.37±0.2 | 80.7±1.5 | 49.3±3.0 | 33.0±2.7 | 4.24±0.1 | 86.7±0.6 | 63.4±1.5 | 52.3±1.3 | 19.6±0.7 |
| | | 6 bit | 56.5±0.4 | 15.3±0.5 | 15.2±0.2 | 1.30±0.1 | 64.0±3.8 | 26.6±1.1 | 18.7±0.5 | 1.71±0.3 | 84.2±1.0 | 51.4±2.2 | 40.0±3.0 | 4.89±0.2 |
| | | 32 bit | 80.2±2.6 | 43.1±4.3 | 30.0±4.8 | 15.6±1.3 | 83.7±0.8 | 54.1±2.3 | 45.3±2.3 | 29.8±0.3 | 90.3±0.3 | 72.4±1.0 | 57.3±0.3 | 40.7±0.6 |
| FedFTG | w avg | 8 bit | 82.0±1.1 | 52.8±0.7 | 39.6±0.9 | 22.9±0.8 | 87.7±0.2 | 59.1±0.2 | 51.2±0.3 | 35.8±0.2 | 90.9±0.3 | 73.7±0.9 | 58.7±1.7 | 42.9±0.3 |
| | | 6 bit | 80.0±0.4 | 50.6±0.5 | 39.6±0.4 | 14.3±0.2 | 81.9±0.3 | 55.7±0.4 | 45.0±0.3 | 17.5±0.2 | 85.2±0.1 | 60.5±0.1 | 48.6±0.1 | 25.8±0.1 |
| | | 32 bit | 82.0±0.7 | 52.3±0.7 | 43.0±0.5 | 21.0±1.4 | 87.6±0.6 | 57.7±0.4 | 51.3±0.1 | 31.9±0.3 | 91.0±0.2 | 73.0±1.1 | 59.3±0.9 | 42.1±0.3 |
| | w/o avg | 8 bit | 74.4±1.4 | 26.7±1.4 | 22.2±1.9 | 6.5±0.6 | 79.6±1.1 | 37.8±1.3 | 37.1±3.0 | 11.2±0.5 | 89.1±0.8 | 63.0±1.1 | 48.5±1.5 | 24.7±0.4 |
| | | 6 bit | 65.2±2.9 | 24.9±1.4 | 17.0±1.5 | 2.3±0.7 | 73.5±2.1 | 31.0±2.1 | 25.8±2.3 | 6.2±0.2 | 83.1±0.9 | 48.2±0.9 | 39.1±2.5 | 10.2±0.4 |
| | | 32 bit | 80.6±2.2 | 46.1±7.3 | 32.4±1.0 | 15.9±1.9 | 84.6±1.7 | 58.9±4.1 | 47.3±1.9 | 28.3±0.4 | 90.7±0.4 | 73.8±1.0 | 57.9±0.3 | 41.6±0.6 |

We also place the algorithms used here, and these pseudocodes are added to our methods. When we conduct experiments, we also follow the algorithm flow from Algorithm 3 to Algorithm 6. We found that although different FL methods will behave differently when the client updates or the server updates, our method can be easily applied to them.

## E  Efficient FL Detail

We list the detailed results of the Efficient FL used in experiments. We use the same network structure ConvNet and the CIFAR10 split in $\alpha = 0.01$ to conduct experiments. In HeteroFL, a, b, c, d, and e respectively represent $100\%$, $50\%$, $25\%$, $12.5\%$, and $6.25\%$ channel proportions. In Table 5, for example, a1-b1-c1 means that during local training, a, b, c, the three models account for 1:1:1. In SplitMix, a, b, c, d, e, have the same representation. 1/8 and 1/16 of SplitMix represent the minimum model channel ratio after segmentation. We conducted experiments on 1/8 and 1/16 respectively.

We calculated the average memory consumption of a single device during each communication round, for each compression method, and also for our method. For memory consumption statistics, we

record model size, gradient storage, and intermediate activation storage. Table 5 to Table 8 contains all our experimental results comparing memory savings.

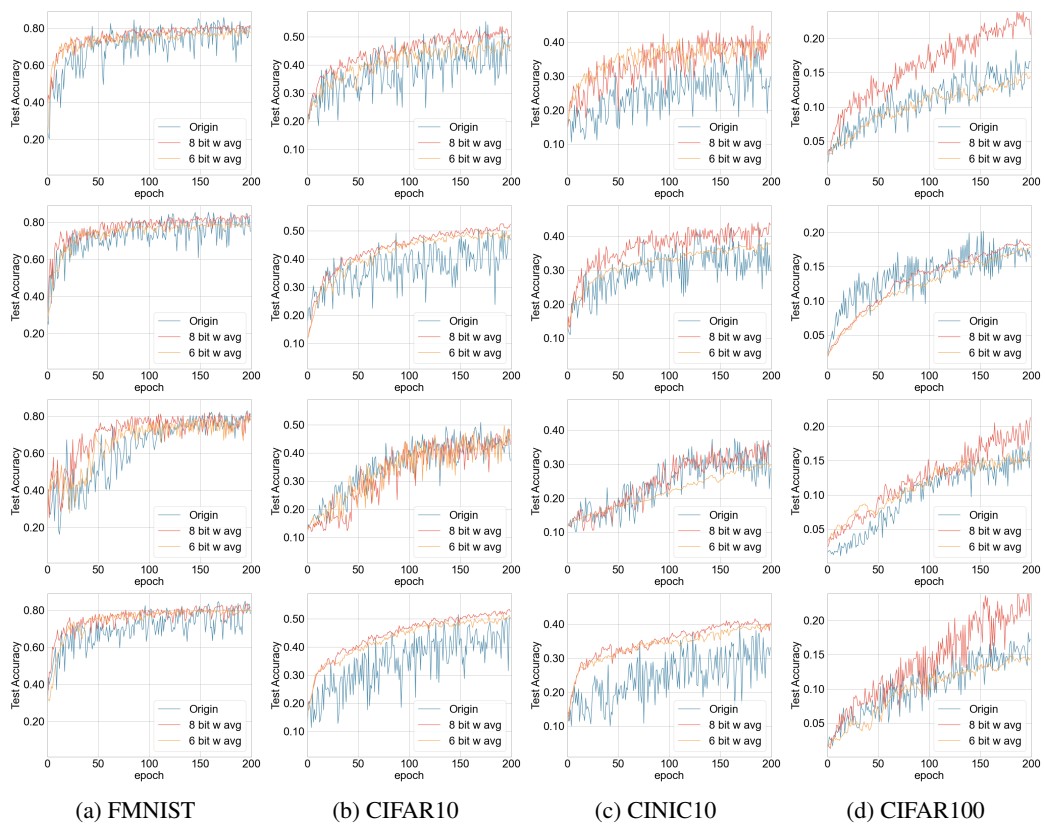

(a) FMNIST      (b) CIFAR10      (c) CINIC10      (d) CIFAR100

Figure 4: Visualization of the accuracy of ABAvg, FedProx, FedGen, FedFTG FL methods when $\alpha = 0.01$ on 4 datasets. The picture from top to bottom is ABAvg, FedProx, FedGen, FedFTG.

Table 5: HeteroFL. a1, b1, c1, d1, e1 represent the percentage of the number of model channels to the original number: 1, 0.5, 0.25, 0.125, 0.625. For example, a1-b1-c1 means that the a, b, and c models exist in a 1:1:1 ratio.

| Model | Sparsity | Accuracy (%) | Memory (MB) |
|---|---|---|---|
| a1 | 0.00 | $37.63 \pm 1.37$ | 205.4 |
| a1-b1 | 0.25 | $26.53 \pm 2.34$ | 144.5 |
| a1-b1-c1 | 0.41 | $28.78 \pm 1.44$ | 109.6 |
| a1-b1-c1-d1 | 0.53 | $34.45 \pm 0.98$ | 87.8 |
| a1-b1-c1-d1-e1 | 0.61 | $25.92 \pm 1.79$ | 73.1 |
| b1 | 0.50 | $36.36 \pm 0.47$ | 83.7 |
| b1-c1 | 0.62 | $37.84 \pm 1.85$ | 61.8 |
| b1-c1-d1 | 0.70 | $21.78 \pm 1.53$ | 48.6 |
| b1-c1-d1-e1 | 0.76 | $26.35 \pm 1.60$ | 40.1 |
| c1 | 0.75 | $33.05 \pm 1.24$ | 39.9 |
| c1-d1 | 0.81 | $29.19 \pm 2.12$ | 31.1 |
| c1-d1-e1 | 0.85 | $31.55 \pm 1.56$ | 25.5 |
| d1 | 0.87 | $19.61 \pm 1.35$ | 22.3 |
| d1-e1 | 0.90 | $27.95 \pm 0.77$ | 18.4 |
| e1 | 0.93 | $26.20 \pm 0.22$ | 14.5 |

Table 6: SplitMix 1/8. Set the smallest model to 1/8 of the original model, which is the d model. By setting the client's resource limit, for example, a1-b1-c1, the client's resource limit is 1, 1/2, 1/4, then 8, 4, and 2 d models can be trained in parallel respectively.

| Model | Sparsity | Accuracy (%) | Memory (MB) |
|---|---|---|---|
| a1 | 0.00 | $41.04 \pm 2.31$ | 178.4 |
| a1-b1 | 0.25 | $35.24 \pm 5.35$ | 133.8 |
| a1-b1-c1 | 0.41 | $32.84 \pm 2.50$ | 104.1 |
| a1-b1-c1-d1 | 0.53 | $31.19 \pm 3.17$ | 83.6 |
| b1 | 0.50 | $40.36 \pm 3.56$ | 89.2 |
| b1-c1 | 0.62 | $38.31 \pm 2.76$ | 66.9 |
| b1-c1-d1 | 0.70 | $30.25 \pm 4.48$ | 52.0 |
| c1 | 0.75 | $39.18 \pm 4.98$ | 44.6 |
| c1-d1 | 0.81 | $38.89 \pm 3.50$ | 33.4 |
| d1 | 0.87 | $41.80 \pm 2.40$ | 22.3 |

Table 7: SplitMix 1/16. Set the smallest model to 1/16 of the original model, which is the e model. By setting the client's resource limit, for example, a1-b1-c1, the client's resource limit is 1, 1/2, 1/4, then 16, 8, and 4 e models can be trained in parallel respectively.

| Model | Sparsity | Accuracy (%) | Memory (MB) |
|---|---|---|---|
| a1 | 0.00 | $30.72 \pm 3.61$ | 232.0 |
| a1-b1 | 0.25 | $29.62 \pm 2.01$ | 174.0 |
| a1-b1-c1 | 0.41 | $27.75 \pm 2.92$ | 135.3 |
| a1-b1-c1-d1 | 0.53 | $32.08 \pm 4.14$ | 108.8 |
| a1-b1-c1-d1-e1 | 0.61 | $27.09 \pm 5.31$ | 89.9 |
| b1 | 0.50 | $33.73 \pm 3.52$ | 116.0 |
| b1-c1 | 0.62 | $32.77 \pm 4.44$ | 87.0 |
| b1-c1-d1 | 0.70 | $25.60 \pm 2.81$ | 67.7 |
| b1-c1-d1-e1 | 0.76 | $26.93 \pm 2.70$ | 54.4 |
| c1 | 0.75 | $35.12 \pm 2.00$ | 58.0 |
| c1-d1 | 0.81 | $32.53 \pm 2.29$ | 43.5 |
| c1-d1-e1 | 0.85 | $27.33 \pm 2.53$ | 33.8 |
| d1 | 0.87 | $37.33 \pm 2.27$ | 29.0 |
| d1-e1 | 0.90 | $34.04 \pm 2.68$ | 21.8 |
| e1 | 0.93 | $31.72 \pm 2.15$ | 14.5 |

Table 8: Low precision FL. We tested the accuracy at different precisions and calculated the memory usage during local training.

| Precision | Accuracy (%) | Memory (MB) |
|---|---|---|
| 8 bit | $53.32 \pm 2.78$ | 51.3 |
| 6 bit | $49.37 \pm 0.95$ | 38.5 |
| 5 bit | $44.67 \pm 1.04$ | 32.1 |

## F   Limitations

In fact, what we performed in the experiment was fake quantization, that is, simulated quantization, which means we only focused on the impact of quantization on accuracy. The actual quantization operation requires the cooperation of hardware, and the hardware GPU we used was GeForce RTX 4090, which could not directly perform low precision training, and could not reflect the acceleration effect of our method.

# G   Broader Impacts

FL, while having a positive impact in areas such as data privacy protection and cross-domain collaboration, also faces some potential negative impacts and challenges.

First, in FL, attackers might attempt to implant backdoors during the model training process. By introducing specific triggers into the training data, attackers can activate the backdoor after the model is deployed, causing the model to output results that the attacker desires. This type of attack poses a serious threat to the model's security and needs to be guarded against with strict security measures and auditing processes.

Second, attackers could disrupt the performance of the model by injecting incorrect information into the training data. This data poisoning can occur at the client side or during the model aggregation process. Data poisoning attacks may lead to the model behaving abnormally under certain conditions, affecting the model's reliability and accuracy.

Last, although FL is designed to protect user privacy by training models locally and sharing only model updates rather than raw data, this method is not foolproof. Attackers might infer sensitive information about the training data by analyzing the model's update information, such as gradients or weight changes. This kind of privacy leakage can be mitigated with techniques like differential privacy, which may impact the model's performance.

---

**Algorithm 3** Low Precision FL with ABAvg

---

**Input:** Quantization functions $Q_A, Q_E, Q_G, Q_M, Q_W$; Momentum coefficient $\rho$; L layers DNN $\{f_1, f_2, \ldots, f_L\}$; Loss function $\ell$; Validation dataset $\mathcal{D}_v$.

1: **Initialize:** $\mathbf{w}_0, \bar{\mathbf{w}}_0 \leftarrow \mathbf{w}_0$
2: **for** $t = 0, 1, \ldots, T-1$ **do**
3:     **if** $t \equiv 0 \pmod{E}$ **then**
4:         Select $K$ clients from $[N]$ to be $\mathcal{S}_t$
5:         $\mathbf{w}_t^k \leftarrow Q(\bar{\mathbf{w}}_t), k \in \mathcal{S}_t$
6:     **end if**
7:     **for** $k \in \mathcal{S}_t$ **do**
8:         $\mathbf{w}_{t+1}^k \leftarrow$ **ClientUpdate**$(t, k, \mathbf{w}_t^k)$
9:     **end for**
10:    **if** $t + 1 \in \mathcal{I}_E$ **then**
11:        Get $a_k$ from testing accuracy of each client $k \in \mathcal{S}_{t'}$ on $\mathcal{D}_v$
12:        $p_k = \frac{a_k}{\sum_{i \in \mathcal{S}_{t'}} a_i}$
13:        $\mathbf{w}_{t+1} \leftarrow \sum_{k \in \mathcal{S}_{t'}} p_k \mathbf{w}_{t+1}^k$
14:        $\bar{\mathbf{w}}_{t+1} \leftarrow \lambda \bar{\mathbf{w}}_{t'} + (1 - \lambda)\mathbf{w}_{t+1}$
15:    **end if**
16: **end for**
17: **Return:** $\bar{\mathbf{w}}_T$
18:
19: **ClientUpdate**$(t, k, w_t^k)$:
20:    Get batch $(x_{k,j_t}, y_{k,j_t})$ from $\mathcal{D}_k$
21:    **Forward Propagation:**
22:        $(a_t^k)^{(0)} = x_{k,j_t}$
23:        $(a_t^k)^{(l)} = Q_A(f_l((a_t^k)^{(l-1)}, (w_t^k)^{(l)})), \forall l \in [1, L]$
24:    **Backward Propagation:**
25:        $(e_t^k)^{(L)} = \nabla_{(a_t^k)^{(L)}} \ell((a_t^k)^{(L)}, y_{k,j_t})$
26:        $(e_t^k)^{(l-1)} = Q_E(\frac{\partial f_l((a_t^k)^{(l-1)}, (w_t^k)^{(l)})}{\partial (a_t^k)^{(l-1)}}(e_t^k)^{(l)}), \forall l \in [1, L]$
27:        $(g_t^k)^{(l)} = Q_G(\frac{\partial f_l((a_t^k)^{(l-1)}, (w_t^k)^{(l)})}{\partial (w_t^k)^{(l)}}(e_t^k)^{(l)}), \forall l \in [1, L]$
28:    **Low Precision SGD Update:**
29:        $(v_{t+1}^k)^{(l)} \leftarrow Q_M(\rho(v_t^k)^{(l)} + (g_t^k)^{(l)}), \forall l \in [1, L]$
30:        $(w_{t+1}^k)^{(l)} \leftarrow Q_W((w_t^k)^{(l)} - \eta_t \cdot (v_{t+1}^k)^{(l)}), \forall l \in [1, L]$
31:    **Return:** $w_{t+1}^k$

---

**Algorithm 4** Low Precision FL with FedProx

---

**Input:** Quantization functions $Q_A, Q_E, Q_G, Q_M, Q_W$; Momentum coefficient $\rho$; L layers DNN $\{f_1, f_2, \ldots, f_L\}$; Loss function $\ell$; FedProx proximal term $\mu$.

1: **Initialize:** $\mathbf{w}_0, \bar{\mathbf{w}}_0 \leftarrow \mathbf{w}_0$
2: **for** $t = 0, 1, \ldots, T - 1$ **do**
3:    **if** $t \equiv 0 \pmod{E}$ **then**
4:       Select $K$ clients from $[N]$ to be $\mathcal{S}_t$
5:       $\mathbf{w}_t^k \leftarrow Q(\bar{\mathbf{w}}_t), k \in \mathcal{S}_t$
6:    **end if**
7:    **for** $k \in \mathcal{S}_t$ **do**
8:       $\mathbf{w}_{t+1}^k \leftarrow \textbf{ClientUpdate}(t, k, \mathbf{w}_{\lfloor \frac{t}{E} \rfloor E}^k, \mathbf{w}_t^k)$
9:    **end for**
10:   **if** $t + 1 \in \mathcal{I}_E$ **then**
11:      $\mathbf{w}_{t+1} \leftarrow \sum_{k \in \mathcal{S}_{t'}} \frac{p_k}{q_{t'}} \mathbf{w}_{t+1}^k$
12:      $\bar{\mathbf{w}}_{t+1} \leftarrow \lambda \bar{\mathbf{w}}_{t'} + (1 - \lambda)\mathbf{w}_{t+1}$
13:   **end if**
14: **end for**
15: **Return:** $\bar{\mathbf{w}}_T$
16:
17: **ClientUpdate**$(t, k, w_{\lfloor \frac{t}{E} \rfloor E}^k, w_t^k)$**:**
18:   FedProx loss function $\mathcal{H} = \ell + \frac{\mu}{2}||w_{\lfloor \frac{t}{E} \rfloor E}^k - w_t^k||^2$
19:   Get batch $(x_{k,j_t}, y_{k,j_t})$ from $\mathcal{D}_k$
20:   **Forward Propagation:**
21:     $(a_t^k)^{(0)} = x_{k,j_t}$
22:     $(a_t^k)^{(l)} = Q_A(f_l((a_t^k)^{(l-1)}, (w_t^k)^{(l)})), \forall l \in [1, L]$
23:   **Backward Propagation:**
24:     $(e_t^k)^{(L)} = \nabla_{(a_t^k)^{(L)}} \ell((a_t^k)^{(L)}, y_{k,j_t})$
25:     $(e_t^k)^{(l-1)} = Q_E(\frac{\partial f_l((a_t^k)^{(l-1)}, (w_t^k)^{(l)})}{\partial (a_t^k)^{(l-1)}} (e_t^k)^{(l)}), \forall l \in [1, L]$
26:     $(g_t^k)^{(l)} = Q_G(\frac{\partial f_l((a_t^k)^{(l-1)}, (w_t^k)^{(l)})}{\partial (w_t^k)^{(l)}} (e_t^k)^{(l)} + \mu((w_t^k)^{(l)} - (w_{\lfloor \frac{t}{E} \rfloor E}^k)^{(l)})), \forall l \in [1, L]$
27:   **Low Precision SGD Update:**
28:     $(v_{t+1}^k)^{(l)} \leftarrow Q_M(\rho(v_t^k)^{(l)} + (g_t^k)^{(l)}), \forall l \in [1, L]$
29:     $(w_{t+1}^k)^{(l)} \leftarrow Q_W((w_t^k)^{(l)} - \eta_t \cdot (v_{t+1}^k)^{(l)}), \forall l \in [1, L]$
30:   **Return:** $w_{t+1}^k$

---

**Algorithm 5** Low Precision FL with FedGen

---

**Input:** Quantization functions $Q_A, Q_E, Q_G, Q_M, Q_W$; Momentum coefficient $\rho$; L layers DNN $\{f_1, f_2, \ldots, f_L\}$; Loss function $\ell$; Generator parameter $\theta$; $\hat{p}(y)$ uniformly initialized; local label counter $c_k$.

1: **Initialize:** $\mathbf{w}_0, \bar{\mathbf{w}}_0 \leftarrow \mathbf{w}_0$
2: **for** $t = 0, 1, \ldots, T - 1$ **do**
3:     **if** $t \equiv 0 \pmod E$ **then**
4:         Select $K$ clients from $[N]$ to be $\mathcal{S}_t$
5:         Update $c_k, k \in \mathcal{S}_t$
6:         $\mathbf{w}_t^k \leftarrow Q(\bar{\mathbf{w}}_t), k \in \mathcal{S}_t$
7:     **end if**
8:     **for** $k \in \mathcal{S}_t$ **do**
9:         $\mathbf{w}_{t+1}^k \leftarrow$ **ClientUpdate**$(t, k, \mathbf{w}_t^k, \hat{p}(y), \theta)$
10:     **end for**
11:     **if** $t + 1 \in \mathcal{I}_E$ **then**
12:         $\mathbf{w}_{t+1} \leftarrow \sum_{k \in \mathcal{S}_{t'}} \frac{p_k}{q_{t'}} \mathbf{w}_{t+1}^k$
13:         $\bar{\mathbf{w}}_{t+1} \leftarrow \lambda \bar{\mathbf{w}}_{t'} + (1 - \lambda)\mathbf{w}_{t+1}$
14:         Server updates $\hat{p}(y)$ based on $\{c_k\}_{k \in \mathcal{S}_{t'}}$
15:         Generator updates $\theta = \underset{\theta}{\arg\min}\, \mathbb{E}_{y \sim \hat{p}(y)} \mathbb{E}_{z \sim G_\theta(z|y)} [\ell(\frac{1}{K} \sum_{k \in \mathcal{S}_{t'}} f_L(z, (\mathbf{w}_{t+1})^{(L)}), y)]$
16:     **end if**
17: **end for**
18: **Return:** $\bar{\mathbf{w}}_T$
19:
20: **ClientUpdate**$(t, k, w_t^k, \hat{p}(y), \theta)$**:**
21:     Get batch $(x_{k,j_t}, y_{k,j_t})$ from $\mathcal{D}_k, \hat{y}_t^k \sim \hat{p}(y), \hat{z}_t^k \sim G_\theta(\cdot|\hat{y}_t^k)$
22:     **Forward Propagation:**
23:         $(a_t^k)^{(0)} = x_{k,j_t}$
24:         $(a_t^k)^{(l)} = Q_A(f_l((a_t^k)^{(l-1)}, (w_t^k)^{(l)})), \forall l \in [1, L]$
25:     **Backward Propagation:**
26:         $(e_t^k)^{(L)} = \nabla_{(a_t^k)^{(L)}} \ell((a_t^k)^{(L)}, y_{k,j_t})$
27:         $(e_t^k)^{(l-1)} = Q_E(\frac{\partial f_l((a_t^k)^{(l-1)}, (w_t^k)^{(l)})}{\partial (a_t^k)^{(l-1)}} (e_t^k)^{(l)}), \forall l \in [1, L]$
28:         $(g_t^k)^{(l)} = Q_G(\frac{\partial f_l((a_t^k)^{(l-1)}, (w_t^k)^{(l)})}{\partial (w_t^k)^{(l)}} (e_t^k)^{(l)} + \frac{\partial \ell(f_L(\hat{z}_t^k, (w_t^k)^{(L)}), \hat{y}_t^k)}{\partial (w_t^k)^{(L)}}), \forall l \in [1, L]$
29:     **Low Precision SGD Update:**
30:         $(v_{t+1}^k)^{(l)} \leftarrow Q_M(\rho(v_t^k)^{(l)} + (g_t^k)^{(l)}), \forall l \in [1, L]$
31:         $(w_{t+1}^k)^{(l)} \leftarrow Q_W((w_t^k)^{(l)} - \eta_t \cdot (v_{t+1}^k)^{(l)}), \forall l \in [1, L]$
32:     **Return:** $w_{t+1}^k$

---

**Algorithm 6** Low Precision FL with FedFTG

**Input:** Quantization functions $Q_A, Q_E, Q_G, Q_M, Q_W$; Momentum coefficient $\rho$; L layers DNN $\{f_1, f_2, \ldots, f_L\}$; Loss function $\ell$; Generator parameter $\theta$; Server training iteration $I$; Inner training iteration of the generator and the server $I_g, I_d$; $\ell_{md}, \ell_{cls}, \ell_{dis}$ is the loss used in ([40]).

1: **Initialize:** $\mathbf{w}_0, \bar{\mathbf{w}}_0 \leftarrow \mathbf{w}_0$
2: **for** $t = 0, 1, \ldots, T-1$ **do**
3:    **if** $t \equiv 0 \pmod{E}$ **then**
4:       Select $K$ clients from $[N]$ to be $\mathcal{S}_t$
5:       $\mathbf{w}_t^k \leftarrow Q(\bar{\mathbf{w}}_t), k \in \mathcal{S}_t$
6:    **end if**
7:    **for** $k \in \mathcal{S}_t$ **do**
8:       $\mathbf{w}_{t+1}^k \leftarrow$ **ClientUpdate**$(t, k, \mathbf{w}_t^k)$
9:    **end for**
10:   **if** $t + 1 \in \mathcal{I}_E$ **then**
11:      $\mathbf{w}_{t+1} \leftarrow$ **ServerUpdate**$(\theta, \{\mathbf{w}_{t+1}^k\}_{k \in \mathcal{S}_{t'}})$
12:      $\bar{\mathbf{w}}_{t+1} \leftarrow \lambda \bar{\mathbf{w}}_{t'} + (1 - \lambda)\mathbf{w}_{t+1}$
13:   **end if**
14: **end for**
15: **Return:** $\bar{\mathbf{w}}_T$
16:
17: **ClientUpdate**$(t, k, w_t^k)$**:**
18:   Get batch $(x_{k,j_t}, y_{k,j_t})$ from $\mathcal{D}_k$
19:   **Forward Propagation:**
20:     $(a_t^k)^{(0)} = x_{k,j_t}$
21:     $(a_t^k)^{(l)} = Q_A(f_l((a_t^k)^{(l-1)}, (w_t^k)^{(l)})), \forall l \in [1, L]$
22:   **Backward Propagation:**
23:     $(e_t^k)^{(L)} = \nabla_{(a_t^k)^{(L)}} \ell((a_t^k)^{(L)}, y_{k,j_t})$
24:     $(e_t^k)^{(l-1)} = Q_E\left(\frac{\partial f_l((a_t^k)^{(l-1)}, (w_t^k)^{(l)})}{\partial (a_t^k)^{(l-1)}}(e_t^k)^{(l)}\right), \forall l \in [1, L]$
25:     $(g_t^k)^{(l)} = Q_G\left(\frac{\partial f_l((a_t^k)^{(l-1)}, (w_t^k)^{(l)})}{\partial (w_t^k)^{(l)}}(e_t^k)^{(l)}\right), \forall l \in [1, L]$
26:   **Low Precision SGD Update:**
27:     $(v_{t+1}^k)^{(l)} \leftarrow Q_M(\rho(v_t^k)^{(l)} + (g_t^k)^{(l)}), \forall l \in [1, L]$
28:     $(w_{t+1}^k)^{(l)} \leftarrow Q_W((w_t^k)^{(l)} - \eta_t \cdot (v_{t+1}^k)^{(l)}), \forall l \in [1, L]$
29: **Return** $w_{t+1}^k$
30:
31: **ServerUpdate**$(\theta, \{\mathbf{w}_{t+1}^k\}_{k \in \mathcal{S}_{t'}})$**:**
32:   $\mathbf{w}_{t+1} = \sum_{k \in \mathcal{S}_{t'}} \frac{p_k}{q_{t'}} \mathbf{w}_{t+1}^k$
33:   Compute $p_{t'}(y) \propto \sum_{k \in \mathcal{S}_{t'}} \sum_{j=1}^{|\mathcal{D}_k|} \mathbb{E}_{(x_{k,j}, y_{k,j}) \sim \mathcal{D}_k}\left[1_{y_{k,j}=y}\right] = \sum_{k \in \mathcal{S}_{t'}} n_k^y$
34:   **for** $i = 1, 2, \ldots, I$ **do**
35:     Get batch $(Z, Y)$ from $z \sim \mathcal{N}(0, 1)$ and $y \sim p_{t'}(y)$
36:     **for** $j = 1, 2, \ldots, I_g$ **do**
37:       Update $\theta$ according to $\min_{\mathbf{w}_{t+1}} \max_{\theta} \mathbb{E}_{z \sim \mathcal{N}(\mathbf{0},1), y \sim p_{t'}(y)}\left[\ell_{md} - \lambda_{cls}\ell_{cls} - \lambda_{dis}\ell_{dis}\right]$
38:     **end for**
39:     **for** $j = 1, 2, \ldots, I_d$ **do**
40:       Update $\mathbf{w}_{t+1}$ according to $\min_{\mathbf{w}_{t+1}} \max \theta \mathbb{E}_{z \sim \mathcal{N}(\mathbf{0},1), y \sim p_{t'}(y)}\left[\ell_{md} - \lambda_{cls}\ell_{cls} - \lambda_{dis}\ell_{dis}\right]$
41:     **end for**
42:   **end for**
43:   **Return** $\mathbf{w}_{t+1}$

