# OpenReview forum: "Low Precision Local Training is Enough for Federated Learning"
_NeurIPS.cc/2024/Conference — NeurIPS 2024 poster_

### Official Review · Reviewer_1d4C · 2024-06-16

**Soundness:** 3
**Presentation:** 3
**Contribution:** 3
**Rating:** 6
**Confidence:** 4

**Summary:**

This paper proposes an efficient federated learning (FL) paradigm, where the local models in the clients are trained with low-precision operations and communicated with the server in low precision format, while only the model aggregation in the server is performed with high-precision computation. The performance is comparable to full-precision training, and sometimes even better since the over-fitting issue in local training is relieved.

**Strengths:**

S1. The idea of applying SWALP's low-precision training within each cycle for local training in FL is meaningful and effective.
S2. There are theoretical analysis on convergence.
S3. Experiments are comprehensive and demonstrate the effectiveness of the proposed method.

**Weaknesses:**

W1. The biggest concern is regarding the novelty w.r.t. SWALP [40]. The entire Sec 3.2 and Sec 4.1 are almost the same as in [40]. The only difference seems to be Sec 4.2 but still very similar to the idea of SWA, just the setting of aggregation changes from by cycles to by clients, and a moving average of parameters is used.

W2. Writing needs improvement. For example, there is a typo "sever" in Line 108.
Eq (5) is different from its counterpart in [40] where the power was F-1 but now W-F-1. please explain the reason of difference.
Line 135, missing a space before "i.e."
Eq(7) uses E which is not clear until continuing reading to Line 161 and Algorithm 2 Lines 3-4.
Algorithm 2 Line 11, t' is only briefly mentioned in Line 161 without even referring to the used lines.

W3. It would be good to estimate the time reduction with the professional hardware (real acceleration).

**Questions:**

Justify method novelty against [40] (W1) and answer related questions in W2.

---

> ### Author Rebuttal · Authors · 2024-08-07
>
> **Q1: The biggest concern is regarding the novelty w.r.t. SWALP [40].**
>
> **A1:** We summarize the differences between our method and SWALP as follows:
>
> 1) SWALP is designed for standard sgd in centralized training, our approach focuses on FL.
> 2) We show both empirically and theoretically the effecitivenss of our approach in heterogeneous datasets.
> 3) We show that besides FedAVG, our method can be integrated with various FL algorithms.
>
>
> **Q2: Typos, e.g., "sever" in Line 108 and missing a space before "i.e.".**
>
> **A2:** Thanks. We will correct the typos and improve the writing accordingly.
>
> **Q3: Eq (5) is different from its counterpart in [40] where the power was F-1 but now W-F-1. Please explain the reason of difference.**
>
> **A3:** The reason for this issue is that in Section 3.1 of SWALP, the definitions of W and F are different between the "Fixed Point Quantization" section and the "Block Floating Point (BFP) Quantization" section. Our definition of W and F is the same as in the "Fixed Point Quantization" section. In our paper and "Fixed Point Quantization" section, F represents the number of bits occupied by the fractional part, while in the "Block Floating Point (BFP) Quantization" section, F stands for the number of bits occupied by the shared exponent.
>
>
> **Q4: Eq(7) uses E which is not clear until continuing reading to Line 161 and Algorithm 2 Lines 3-4.**
>
> **A4:** We provided the definition of E in line 115 of the paper.
>
> **Q5: Algorithm 2 Line 11, t' is only briefly mentioned in Line 161 without even referring to the used lines.**
>
> **A5:** Thank you for your suggestion, we will add further description after t'.
>
> **Q6: It would be good to estimate the time reduction with the professional hardware (real acceleration).**
>
> **A6:** It is difficult to conduct experiments on professional hardware in university, which is expensive and involves hardware programming. Fortunately, we note that our simulation is standard and approved by the community. The reasons are
> 1) Such low precision training methods can be implemented on the machine learning accelerators efficiently to achieve real saving in computational and communication cost;
> 2) The results would be always consistent with those obtained from simulation.
>
> For details, please refer to [r9] and [r10].
>
>
>
> [r9] Kuzmin A, et al. Fp8 Quantization: The Power of the Exponent. NeurIPS 2022.
>
> [r10] Nagel M, et al. Overcoming Oscillations in Quantization-Aware Training. ICML 2022.

---

> > ### Comment · Reviewer_1d4C · 2024-08-12
> >
> > Thanks for the responses. I still feel the method a bit similar to SWALP, and writing needs some work to be clear.
> > I will keep my rating since it was very positive already.

---

> ### Author Response · Authors · 2024-08-13
>
> Dear Reviewer 1d4C,
>
> We sincerely appreciate your support and valuable comments on improving our paper. We believe the primary similarity between SWALP and our method is their high-level inspiration from Kolmogorov’s law, which asserts that the sample average can almost surely converge to the expected value despite the presence of noise. We also find it highly appropriate to develop efficient low-precision local training methods for federated learning, given that the clients in many federated learning applications are resource-constrained. Our proposed method is both simple and effective, presenting a new framework/paradigm for accelerating federated learning. We hope it will inspire researchers to create more efficient federated learning algorithms based on low-precision local training in the future. We believe this contribution holds comparable significance to our proposed method itself.
>
> As the deadline for the author-reviewer discussion phase approaches, could you please let us know if any further discussion or clarification is needed? Thank you again for your valuable time; your support is greatly appreciated.
>
> Sincerely,
> Authors of Submission 11555

---

### Official Review · Reviewer_vykk · 2024-07-10

**Soundness:** 3
**Presentation:** 3
**Contribution:** 1
**Rating:** 5
**Confidence:** 4

**Summary:**

The paper proposes a federated learning approach that performs local training on low precision through quantization combined with a high-precision averaging and a moving average at the server. The paper guarantees convergence and empirically compares several levels of low-precision local training to full-precision training on 4 baseline FL methods. It remains unclear, though, what the contribution of the method is: the main focus seems to be on performance in terms of test accuracy, but the experiments do not show a significant improvement over existing methods. The method supposedly improves communication and computation efficiency but is not empirically compared to state-of-the-art methods, such as [1,2,3].

In their rebuttal, the authors provided novel results that address my concerns about missing baselines. While I remain concerned about the limited novelty and the presentation, I believe that the authors will be able to address these issues to some extent in the next version of the manuscript. Therefore, I have decided to increase my score.

\
[1] Liu, Shiyu, et al. "Quantized SGD in Federated Learning: Communication, Optimization and Generalization." International Conference on Neural Information Processing. Singapore: Springer Nature Singapore, 2023.

[2] Kamp, Michael, et al. "Efficient decentralized deep learning by dynamic model averaging." Machine Learning and Knowledge Discovery in Databases: ECML PKDD, 2018.

[3] Reisizadeh, Amirhossein, et al. "Fedpaq: A communication-efficient federated learning method with periodic averaging and quantization." International conference on artificial intelligence and statistics. PMLR, 2020.

**Strengths:**

- convergence guarantees
- clear explanation of the method

**Weaknesses:**

- the presentation should be improved (e.g., several typos, such ass "accept" instead of "aspect", the term "a avg" for "with moving average" is unintuitive)
- the ablation study does not clearly show that low precision local training improves performance, since it is combined with a moving average that has a strong positive impact on performance.
- lack of comparison to baselines
- unclear use-case for the method
- the paper does not discuss existing federated learning with quantization literature in sufficient detail.
- for the non-iid experiments it would be interesting how quantization interacts with local BN layers in FedBN [4]

\
[4] Li, Xiaoxiao, et al. "FedBN: Federated Learning on Non-IID Features via Local Batch Normalization." International Conference on Learning Representations, 2021.

**Questions:**

- It is unclear from the results how much of the benefit stems from quantization and how much from the moving average. Is it correct that the moving average has a very large positive effect regardless of quantization?

**Limitations:**

Section F in the appendix addresses the limitation of simulating quantization. It does not address issues like computation and communication complexity, the applicability of the approach, or limitations in the empirical evaluation.

---

> ### Author Rebuttal · Authors · 2024-08-07
>
> **Q1: Contribution in terms of  test accuracy.**
>
> **A1:** It is a misunderstanding. We do not aim to improve  test accuracy, but rather to demonstrate that low precision local training is sufficient for FL and can be used to reduce training and communication cost. Our method, which performs low precision local training, achieves comparable  (or even higher) accuracy with full-precision methods. The improvements on the training efficiency and communication cost can be guaranteed as our low precision operators are standard.
>
> **Q2: Compare with  FedPAQ, Quantized-SGD and dynamic-model-averaging.**
>
> **A2:** Thanks. In our paper, we gave the results with strong baselines HeteroFL [9] and SplitMix [14]. Below is the result on CIFAR10 ($\alpha=0.01$) of FedPAQ.   "()" denotes the percentage of models on the clients. Following [r6] we use the number of   weights, activation, and gradients of local training to approximate training cost.  It shows that we can reduce communication and computational cost  while maintain high accuracy.
>
> | Method                        | Acc     | Communication cost (MB / round) | Training cost (MB / client) |
> | :-                          | :-:   | :-:                       |:-:                        |
> | HeteroFL (1)                  |41.1±2.4 |   39.1                          |  205.4                      |
> | HeteroFL (1/2)                |36.4±0.5 |   19.6                          |  83.7                       |
> | HeteroFL (1/4,1/8)            |29.2±2.1 |   7.3                           |  31.1                       |
> | SplitMix (1)                  |41.0±2.3 |   39.1                          |  178.4                      |
> | SplitMix (1/2)                |40.4±3.6 |   19.6                          |  89.2                       |
> | SplitMix (1/4,1/8)            |38.9±3.5 |   7.3                           |  33.4                       |
> | FedPAQ 8 bit                  |40.7±2.1 |   9.8                           |  205.4                      |
> | FedPAQ 6 bit                  |38.1±0.7 |   7.3                           |  205.4                      |
> | FedPAQ 5 bit                  |35.0±1.3 |   6.1                           |  205.4                      |
> | FedAVG+Ours(8 bit)                    |53.3±2.8 |   9.8                           |  51.3                       |
> | FedAVG+Ours(6 bit)                    |49.4±1.0 |   7.3                           |  38.5                       |
> | FedAVG+Ours(5 bit)                    |44.7±1.0 |   6.1                           |  32.1                       |
>
> We gave the results of another SOTA CoCoFL in A2 of Reviewer 5cLQ. We have not yet found the code of Quantized-SGD and dynamic-model-averaging, due to the time limitation, we postpone their comparison to revision.
>
> **Q3: Typos.**
>
> **A3:** Thanks. We will improve the writing accordingly.
>
> **Q4: This result  does not clearly show low precision local training improves performance, since it is combined with moving average.**
>
> **A4:** Low-precision training typically tends to decrease instead of improve the accuracy. We introduce it  to reduce communication and computational costs, while  moving average  is adopted to compensate the above accuracy degradation. This point was confirmed in Table 1 and Figure 2. That is that with moving average, the accuracy of the low-precision model can be comparable to full-precision model, and sometimes even better due to its effect on reducing over-fitting risk.
>
> **Q5: Unclear use-case.**
>
> **A5:** Reducing training and communication cost is an important topic in FL since the clients  in FL are always resource-constrained edge devices  with low-bandwidth communication network. We address this challenge by using low precision local training, which can be implemented efficiently on machine learning accelerators. We note that using accelerators with low precision operators is an new paradigm to accelerate AI models. For details, please refer to the surveys [r7] and [r8].
>
> **Q6: Discuss FL with quantization literature.**
>
> **A6:** Thanks. We gave  comparison results with SOTA CoCoFL and FedPAQ with quantization in the table in A2 of Reviewer vykk and the table in A2 of reviewer 5cLQ. We will include them and discuss in details in the revision if accepted.
>
> **Q7: How quantization interacts with  FedBN.**
>
> **A7:** Thanks. The results ($\alpha = 0.01$ ) given below show that our method works well with FedBN. More results will be included in the revision.
>
> | Method        | CIFAR10  | FMNIST   |
> |:-   |:- |:- |
> | FedBN         | 48.7±1.9 | 79.3±0.6 |
> | FedBN+Ours (16 bit) | 49.3±0.4 | 80.2±0.2 |
> | FedBN+Ours (8 bit)  | 48.5±0.3 | 78.6±0.5 |
>
> **Q8: Benefits stem from quantization and moving average.**
>
> **A8:** We would like to clarify  two things:
> 1) Quantization is used to reduce the computational and communication cost and typically it tends to produce negative instead of positive  impact on accuracy and we  adopt  moving average to  alleviate this impact.
> 2) In some tasks, quantization can reduce the risk of over-fitting leading to higher test accuracy. It is verified by Table 1 and  Fig. 2 in our paper.
>
> **Q9: Does moving average have a large positive effect regardless of quantization?**
>
> **A9:**  Moving average is used to reduce the error introduced by quantization. When full precision local training is adopted, its effect is relatively limited compared with that of low-precision, see Table 1 in our paper.
>
> **Q10: More limitation discussion.**
>
> **A10:** Thanks. For computation and communication, we follow the standard settings in FL. We gave a series of additional experimental results in this rebuttal and we will give more discussion in the revision.
>
> [r6] Efficient personalized federated learning via sparse model-adaptation. ICML 2023.
>
> [r7]  AI and ML Accelerator Survey and Trends. IEEE HPEC, 2022.
>
> [r8] Advances and Open Problems in Federated Learning. Foundations and Trends in Machine Learning, 2021.

---

> > ### Comment · Reviewer_vykk · 2024-08-12
> > **Reply to authors**
> >
> > Dear authors,
> >
> > Thank you for your detailed reply and the additional experiments. Please include those points and results in the manuscript. I am leaning towards improving my score after the discussion period with my fellow reviewers.

---

> > > ### Author Response · Authors · 2024-08-13
> > >
> > > Dear Reviewer vykk,
> > >
> > > We sincerely appreciate your response and  final support! Since the deadline for the author-reviewer discussion phase is approaching, would you mind kindly letting us know if any further discussion or clarification is needed? Thank you again for your valuable time, your support is significant to us.
> > >
> > > Sincerely,
> > >
> > > Authors of Submission 11555

---

### Official Review · Reviewer_5cLQ · 2024-07-12

**Soundness:** 3
**Presentation:** 3
**Contribution:** 3
**Rating:** 7
**Confidence:** 4

**Summary:**

The paper studies an FL system with data heterogeneity, a topic has been extensively studied in the past few years. The idea is to perform local training with lower precision through applying block floating point quantization. The idea per se is not new, but proving that the convergence can be achieved using low precision local training is an interesting contribution.

**Strengths:**

The paper is well written and is easy to follow. The idea is also interesting but it is not necessarily new. The most important contribution is the theoretical proof for convergence. The evaluation results in Section 6 validate the theoretical results.

**Weaknesses:**

The paper mainly focuses on data heterogeneity in FL systems. What about resource heterogeneity? It would be important to have a discussion (or better some experimental results) on how the proposed solution perform in such setting. Should we use the same quantization level for all clients, or can we adjust the precision according to the resource availability? Also, the current state of the art of quantization in conjunction with Federated Learning is also missing, e.g.:

- FedQNN[1] uses quantized training in FL.

- CoCoFL[2] uses a combination of quantization and freezing for heterogeneous resources in FL.

How does these SOTA techniques perform compared with the proposed solutions?

[1] Y. Ji and L. Chen, "FedQNN: A Computation–Communication-Efficient Federated Learning Framework for IoT With Low-Bitwidth Neural Network Quantization," in IEEE Internet of Things Journal.

[2] Kilian Pfeiffer, et al. "CoCoFL: Communication-and Computation-Aware Federated Learning via Partial NN Freezing and Quantization." Transactions on Machine Learning Research., 2023

**Questions:**

Please also check my questions in the weakness section.

**Limitations:**

Appendix F clarifies that the fake quantization is applied in the experiments. I really appreciate learning this information, as one of my question was how the quantization is implemented in PyTorch (as PyTorch only supports int8 inference out of the box).
Regarding the broader impact, the discussion is about FL in general. The question is if performing lower precision training at clients improves or worsen these privacy and security aspects (e.g. if data poisoning cab be done easier, as the local updates are low precision).

---

> ### Author Rebuttal · Authors · 2024-08-07
>
> **Q1:  What about resource heterogeneity? It would be important to have a discussion (or better some experimental results) on how the proposed solution perform in such setting. Should we use the same quantization level for all clients, or can we adjust the precision according to the resource availability?**
>
> **A1:** Thanks for your suggestion.  We give the results in the setting of  HeteroFL [9]. That is, we set three types of resource devices: 1, 1/2, and 1/4, which respectively represent the devices that can train the entire network, 1/2 network, and 1/4 network. We use (x, x, x)  to represent the corresponding bits in three type of devices in training. We let $\alpha=0.16$. The experimental results are as follows:
>
> | Precision   | Acc  |
> |:----------  |:---- |
> | (32,32,32)  | 55.6 |
> | (8,8,8)     | 55.4 |
> | (8,8,6)     | 54.7 |
> | (8,6,8)     | 52.6 |
> | (6,8,8)     | 50.4 |
> | (8,6,4)     | 52.8 |
> | (4,6,8)     | 43.7 |
>
> The results above show that our low-precision training mechanism works well in resource heterogeneity applications. They also indicate that when the sum of bit widths used across the three sets of devices is fixed, devices with poorer computational resources should be allocated a smaller number of bit widths.
>
>
> **Q2: How does these SOTA techniques FedQNN and CoCoFL perform compared with the proposed approach?**
>
> **A2:** Thanks for your suggestion. Since the code of FedQNN has not been released,  we give the comparison results with CoCoFL. We follow the setting of CoCoFL and conduct experiments on on Shakespeare and CIFAR10. Following [r3] we calculate the communication cost by counting the size of the model transmitted from the client to the server in each round, and the training cost by counting the weights, activation, and gradients of local training.
>
> | Method      |Shakespeare Acc       | Communication cost (MB / round) | Training cost (MB / client) |
> |:----------  |:---------:            | :-----:                          | :-----:                      |
> | FedAVG      | 49.1                 | 75.5                            | 2927.9                      |
> | CoCoFL      | 49.3                 | 50.3                            | 1951.9                      |
> | FedAVG+Ours(8 bit)| 49.4                 | 18.9                            | 731.9                       |
>
> | Method      |CIFAR10 Acc       | Communication cost (MB / round) | Training cost (MB / client) |
> |:----------  |:---------:        | :-----:                          | :-----:                      |
> | FedAVG      | 84.3             | 21.5                            | 167.7                       |
> | CoCoFL      | 82.0             | 14.3                            | 111.8                       |
> | FedAVG+Ours(8 bit)| 83.1             | 5.4                             | 41.9                        |
>
> The results above show that  our method has a significant reduction in overhead, because our local training is completely low-precision, and the model transmitted to the server is also low-precision. CoCoFL still trains the unfrozen parameters with full precision, and the transmission is also full-precision, so it will have a relatively larger overhead.
>
> **Q3: Appendix F clarifies that the fake quantization is applied in the experiments. I really appreciate learning this information, as one of my question was how the quantization is implemented in PyTorch.**
>
> **A3:** We adopt the widely used simulation method in the studies [40, r4, r5] of low-precision training, which is approved  by the community as its result is always consistent with that on the real ML accelerators. We have submitted our code in the supplementary materials, which includes the specific implementation of fake quantization. Our approach to quantization involves implementing the quantization of weights, activation values, and gradients during the training process. We have three quantization operations: the first is to implement the quantization of activations after every model module, the second is to quantize the gradients of the parameters after the loss is generated and the gradients are backpropagated, the third is to quantize the weights after the optimizer is updated. The specific implementation process can be referred to in our code.
>
>
> **Q4: If performing lower precision training at clients improves or worsen these privacy and security aspects.**
>
> **A4:** We tested different levels of label flip attack on FedAVG and ABAVG [38] by using MNIST, and the results are as follows. ABAVG maintains a validation dataset on the server side and adjusts the weight of the client aggregation according to the client's accuracy, so it can be used for adversarial attacks. We let $\alpha=0.1$. Our experimental results show that low precision training can slightly improve the performance. The reason could be that  low precision training prevents the client from over-fitting  the wrong labeled samples, and thus reduces the impacts of the poisonous data. More results will be included in the revision.
>
> | Method            | Label Flip attack rate |Acc     |
> |:-                 |:-:                      |:-      |
> | FedAVG            | 0.4                    | 60.4   |
> | FedAVG+Ours(8 bit)| 0.4                    | 62.6   |
> | FedAVG            | 0.7                    | 22.8   |
> | FedAVG+Ours(8 bit)| 0.7                    | 27.5   |
> | ABAVG             | 0.4                    | 63.1   |
> | ABAVG+Ours(8 bit)| 0.4                    | 64.0   |
> | ABAVG             | 0.7                    | 37.9   |
> | ABAVG+Ours(8 bit)| 0.7                    | 40.6   |
>
>
>
> [r3] Chen D, et al. Efficient personalized federated learning via sparse model-adaptation. ICML 2023.
>
> [r4] Kuzmin, et al. The Power of the Exponent. NeurIPS 2022.
>
> [r5] Nagel, et al. Overcoming Oscillations in Quantization-Aware Training. ICML 2022.

---

> > ### Comment · Reviewer_5cLQ · 2024-08-09
> >
> > Thanks for the detailed reply. I suggest to integrate the results for Q2 - Q4 in the final version. I have no follow up to these questions. For Q1, I do not see why we need to adapt HeteroFL and adjust the network size at different devices. This could cause some fairness issues, especially when the data is non-i.i.d, as shown in the CoCoFL paper. What if you train the whole model at all devices, but adjust the precision level to the availability of resources at devices?

---

> > > ### Author Response · Authors · 2024-08-10
> > > **Rebuttal by Authors (2)**
> > >
> > > We would like to sincerely thank the reviewer for the thorough and prompt feedback. We appreciate the opportunity to address and clarify the points raised.
> > >
> > > **Q5: What if you train the whole model at all devices, but adjust the precision level to the availability of resources at devices?**
> > >
> > > **A5:** Thanks for your constructive suggestion. Actually, the setting of our previous experiments in the rebuttal is unfair to us rather than to other baselines, as our method is added with much more strict and also unnecessary constraint on the resource.
> > >
> > > According to your advice, below we present the experimental results of training all models by adjusting the precision level  to the availability of resources at devices. We now impose three $\textbf{memory}$ resource constraints (1, 1/2, and 1/4)  on the clients. We would like to clarify that in our previous experiments in the rebuttal,  three types of resource devices: 1, 1/2, and 1/4 respectively represent the devices that can train the entire network, 1/2 channels of the network, and 1/4 channels of the network, which is equivalent to the networks with full, 1/4 and 1/16  $\textbf{memory}$ cost.  Here we use the memory reconstraint and let it to be 1, 1/2 and 1/4 just to make the corresponding precision levels be the widely used ones, i.e., 32, 16 and 8 bits. We list four experimental methods:
> > >     1) FedAVG with full resources, where all the devices have the full resources and be able to train the full network.
> > >     2) HeteroFL adjusts the number of parameters of each client to meet the resource constraints for training.
> > >     3) Our method sets the precision levels of the devices under the three constraints to be 32, 16, and 8 respectively to approximately meet the resource constraints and train the full model.
> > >     4) FedAVG refers to the setting that the network which cannot be trained (due to resource constraints) is directly dropped. We tested the results on CIFAR10, setting $\alpha=0.04$ and $0.16$. The detailed result is given in the table below, which verifies the effectiveness of our method.
> > >
> > >  |Method               |Acc($\alpha=0.04$)|Acc($\alpha=0.16$)|
> > >  |:-|:-:|:-:|
> > > |FedAVG(full resource)|54.4±2.0          |71.9±1.5|
> > > |HeteroFL             |46.3±1.7          |63.4±1.8|
> > > |Ours                 |58.1±0.5          |72.7±0.5|
> > > |FedAVG               |32.4±2.3          |52.8±2.1|

---

> > > > ### Comment · Reviewer_5cLQ · 2024-08-12
> > > >
> > > > Thanks for the new experiments (I was expecting similar results). Please consider including these new experiments in the paper. I confirm my acceptance score.

---

### Official Review · Reviewer_Zy16 · 2024-07-20

**Soundness:** 3
**Presentation:** 3
**Contribution:** 3
**Rating:** 6
**Confidence:** 4

**Summary:**

The paper proposes an efficient Federated Learning (FL) paradigm where local models are trained using low-precision operations and communicated with the central server in low precision format. The aggregation on the server, however, is performed with high-precision computation to ensure accuracy. The authors demonstrate that high-precision models can be recovered from low-precision local models with proper aggregation on the server side. This approach significantly reduces the computational load on client devices and the communication cost. The method is theoretically proven to converge to an optimal solution, even with non-IID data distributions, and extensive experiments show that models trained with low precision (as low as 8 bits) are comparable in performance to those trained with full precision.

**Strengths:**

1. The proposed method reduces the computational and communication overhead for client devices, which is crucial for resource-constrained environments for large models. The paper also provides theoretical guarantees for convergence to the optimal solution, even with non-IID data distributions.
2. The method is effective on the datasets and models in the experiments where low precision training has little to no impact on utility.

**Weaknesses:**

1. The integration of low precision training and high precision aggregation may add complexity to the implementation.The performance improvements are partly dependent on the hardware capabilities, such as the availability of processors supporting low precision operations.
2. The experiments are limited. Only image datasets are considered. Evaluation on other types of data, like text or tabular, can strengthen the results.
3. No integration with differential privacy or other privacy protection mechanisms. Federated learning itself is not private and it would be interesting to see what privacy mechanisms are suitable for low precision model updates.

**Questions:**

1. How can we determine the optimal precision level for local training without extensive hyperparameter tuning which is expensive in FL?
2. Have you explored mixed precision strategy? E.g. using different quantization schema for gradients, activations etc.

**Limitations:**

The authors have discussed the limitations in the paper.

---

> ### Author Rebuttal · Authors · 2024-08-07
>
> **Q1: The integration of low precision training and high precision aggregation may add complexity to the implementation.**
>
> **A1:** Actually, in our method, the transformation between low and high-precision parameters is performed on the server, and the computation in the clients is standard for the clients supporting low precision operators. As we know, in FL, the server always has rich computation and memory resources and the main challenge in improving the training efficiency comes from the resource constrained clients and the communication cost. Therefore, the increased implementation complexity above would not prevent us from improving the overall training efficiency.
>
> **Q2: The performance improvements are partly dependent on the hardware capabilities, such as the availability of processors supporting low precision operations.**
>
> **A2:** Yes. But we would like to point out some promising  progress in this area. In recent years, deep learning applications urge the development of machine learning accelerators. Lots of accelerators supporting low precision operations have been announced, which make it easy to implement the low precision training/inference algorithms including our approach efficiently in practice.  For details, please refer to the survey [r1].
>
>
> **Q3:  Only image datasets are considered. Evaluation on other types of data, like text or tabular, can strengthen the results.**
>
> **A3:** Thanks for your suggestion. Actually, we have conducted experiments on most of the datasets used in existing studies. Below, we give the results on the text dataset IMDB and Shakespeare. The results also confirm our conclusion on text data that low-precision local training is sufficient for Federated Learning. Following [r2], we approximate the communication cost by counting the size of the model transmitted from the client to the server in each round, and the training cost by counting the weights, activation, and gradients of local training.
>
> | Method      |IMDB Acc        | Communication cost (MB / round) | Training cost (MB / client) |
> |:----------  |:---------:      | :-----:                          | :-----:                      |
> | FedAVG      | 75.6           | 323.0                           | 3700.8                      |
> | FedAVG+Ours(8 bit)| 74.8           | 80.8                            | 925.2                       |
>
> | Method      |Shakespeare Acc | Communication cost (MB / round) | Training cost (MB / client) |
> |:----------  |:---------:      | :-----:                          | :-----:                      |
> | FedAVG      | 49.1           | 75.5                            | 2927.9                      |
> | FedAVG+Ours(8 bit)| 49.4           | 18.9                            | 731.9                       |
>
>
> **Q4: No integration with differential privacy or other privacy protection mechanisms. Federated Learning itself is not private and it would be interesting to see what privacy mechanisms are suitable for low precision model updates.**
>
> **A4:** We think that theoretically proving what privacy mechanisms are suitable for low precision model updates is too complicated to be done in the author response period due to the variety of privacy mechanisms. Below, we give the results of FL with differential privacy and low-precision local updates. It shows that our method works well with differential privacy. The reason could be that as our quantization process is stochastic, it can be approximately viewed as a differential privacy protection mechanism and therefore they are compatible with each other.
>
> |Method      | Gaussian Epsilon | Acc  |
> |:-          |:-:                |:-    |
> | FedAVG     | 30               | 45.5 |
> | FedAVG+Ours(8 bit)| 30               | 53.9 |
> | FedAVG     | 10               | 43.6 |
> | FedAVG+Ours(8 bit)| 10               | 53.4 |
> | FedAVG     | 5                | 40.8 |
> | FedAVG+Ours(8 bit)| 5                | 53.0 |
> | FedAVG     | 1                | 38.4 |
> | FedAVG+Ours(8 bit)| 1                | 45.8 |
>
> Moreover, we agree with you that theoretically FL is not private. We give the results with standard FL in the manuscript because FL is still one of the main machine learning paradigms to address the challenges on preserving data privacy. The reason could be that as indicated in Figure 4 of [29] recovering the data with details from the received local models is highly nontrivial.
>
>
> **Q5: How can we determine the optimal precision level for local training without extensive hyperparameter tuning, which is expensive in FL?**
>
> **A5:** Our empirical results show that 8-bit precision level works well on all the tasks. In practice, one can tune the precision level during training, e.g., choose 2 sets of clients with different precision levels in some rounds and choose proper precision level  based on the performance of the two aggregated models.
>
> **Q6: Have you explored mixed precision strategy? E.g. using different quantization schema for gradients, activations etc.**
>
> **A6:** Thanks for your suggestion. We performed local mixed precision training on CIFAR10 with $\alpha=0.01$. We use the triple (x, x, x) to represent the number of bits are used in  quantizing weights, activations, and gradients, respectively. The  results are as follows:
>
> | Mixed precision| Acc      |
> | :--------------|:---------|
> | (32,32,32)     | 52.7±0.6 |
> | (8,8,8)        | 53.3±0.5 |
> | (6,8,8)        | 53.1±0.5 |
> | (8,6,8)        | 50.9±1.0 |
> | (8,8,6)        | 53.2±0.5 |
> | (8,6,4)        | 50.4±1.0 |
> | (6,6,4)        | 50.1±0.7 |
>
> The experimental results indicate that our method also supports mixed-precision. Results with more mixed precision training methods will be included in the revision if accepted.
>
>
>
> [r1] Reuther, et al. AI and ML Accelerator Survey and Trends. IEEE HPEC 2022.
>
> [r2] Chen D, et al. Efficient personalized federated learning via sparse model-adaptation. ICML 2023.

---

> > ### Author Response · Authors · 2024-08-13
> >
> > Dear Reviewer Zy16,
> >
> > We sincerely appreciate for your support and have carefully responded to your constructive comments thoroughly.
> >
> > Since the deadline for the author-reviewer discussion phase is approaching, would you mind kindly letting us know if any further discussion or clarification is needed? Thank you again for your valuable time.
> >
> > Sincerely,
> >
> > Authors of Submission 11555

---

### Decision · Program_Chairs · 2024-09-25

**Decision:**

Accept (poster)

**Comment:**

The reviewers were generally convinced about the communication and computation reduction in the context of FL achieved by the proposed method. They also appreciated both the extent of experiments and the convergence guarantees provided in the paper. On the negative side, there is an overlap with in techniques [40] though, as the authors point out, the application to FL is novel.

Overall, the authors should:
-Include in the paper the additional experiments that they conducted during the rebuttal stage; these can be added in the supplement if there is an issue of space, but hopefully some can surface to the main body of the paper as well.
-Proofread for typos, but also improve presentation w.r.t. aspects indicated by the reviewers.